**Subject Area:**
molecular biology/genomics/cellular biology/bioinformatics/genetics

bio-adhesion, adhesive proteins, glycosylation, transitory adhesion, Patellogastropoda

**Author for correspondence:**
Victor Kang
e-mail: kwk22@cam.ac.uk

# Molecular insights into the powerful mucus-based adhesion of limpets (*Patella vulgata* L.)

Victor Kang[1], Birgit Lengerer[2,3], Ruddy Wattiez[4] and Patrick Flammang[2]

[1]Department of Zoology, University of Cambridge, Cambridge, UK
[2]Biology of Marine Organisms and Biomimetics Unit, Research Institute for Biosciences, University of Mons, Mons 7000, Belgium
[3]Institute of Zoology, University of Innsbruck, 6020 Innsbruck, Austria
[4]Laboratory of Proteomics and Microbiology, Research Institute for Biosciences, University of Mons, Mons 7000, Belgium

VK, 0000-0003-0959-1364; BL, 0000-0002-5431-916X; PF, 0000-0001-9938-1154

Limpets (*Patella vulgata* L.) are renowned for their powerful attachments to rocks on wave-swept seashores. Unlike adult barnacles and mussels, limpets do not adhere permanently; instead, they repeatedly transition between long-term adhesion and locomotive adhesion depending on the tide. Recent studies on the adhesive secretions (bio-adhesives) of marine invertebrates have expanded our knowledge on the composition and function of temporary and permanent bio-adhesives. In comparison, our understanding of the limpets' transitory adhesion remains limited. In this study, we demonstrate that suction is not the primary attachment mechanism in *P. vulgata*; rather, they secrete specialized pedal mucus for glue-like adhesion. Through combined transcriptomics and proteomics, we identified 171 protein sequences from the pedal mucus. Several of these proteins contain conserved domains found in temporary bio-adhesives from sea stars, sea urchins, marine flatworms and sea anemones. Many of these proteins share homology with fibrous gel-forming glycoproteins, including fibrillin, hemolectin and SCO-spondin. Moreover, proteins with potential protein- and glycan-degrading domains could have an immune defence role or assist degrading adhesive mucus to facilitate the transition from stationary to locomotive states. We also discovered glycosylation patterns unique to the pedal mucus, indicating that specific sugars may be involved in transitory adhesion. Our findings elucidate the mechanisms underlying *P. vulgata* adhesion and provide opportunities for future studies on bio-adhesives that form strong attachments and resist degradation until necessary for locomotion.

## 1. Introduction

Limpets (the Patellogastropoda) are an ancient and diverse group of marine gastropods. The earliest fossil records date back to the Middle Ordovician (approx. 450 million years ago), and extant species can be found on seashores around the world [1]. The common limpet, *Patella vulgata* L. (Patellidae), is widespread in Europe and is found in the upper intertidal zone, a challenging habitat with strong forces from tidal waves and currents, as well as prolonged exposure to air and predators [2–4]. Limpets have characteristic conical shells and attach to the surface using their muscular pedal sole. The limpet's powerful attachment is well established, with recorded tenacity values (normal peak attachment force divided by contact area) typically ranging between 0.1 and 0.2 MPa [4–7], reaching 0.7–1.1 MPa in some reports [6,8]. Such impressive attachments help them resist strong tidal waves and thwart predatory attacks [3,9] (figure 1). However, unlike adult mussels and barnacles that rely on

royalsocietypublishing.org/journal/rsob    Open Biol. **10**: 200019

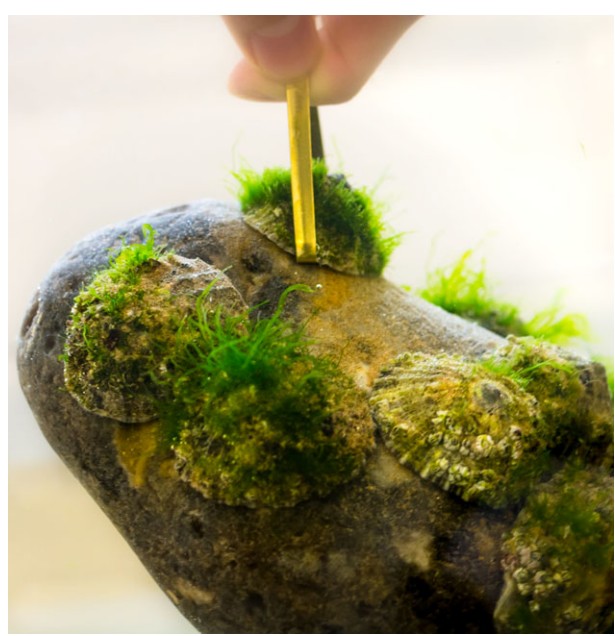

**Figure 1.** Limpets (*Patella vulgata*) have evolved powerful attachments to withstand crashing tidal waves and predatory attacks. Here, one of the authors lifted a heavy rock by hooking onto a single limpet.

filter-feeding and permanently adhere to surfaces in the intertidal zone, limpets are active grazers of biofilm and detritus [10]; hence, they can travel considerable distances while feeding (up to 1.5 m [6]). They must therefore alternate between powerful attachments during stationary periods at low tides and locomotory adhesion at high tides [11]. We refer to this sub-type of transitory adhesion as tidal transitory adhesion.

Despite over a century of research, the mechanisms responsible for the patellid limpets' strong attachment remain unresolved [4]. So far, proposed ideas include: suction [5,11,12] (lowering of pressure beneath the attachment organ by muscle contraction), clamping (muscles forcing shells against the substrate to provide additional friction) [8,13], viscous adhesion (resulting from the flow resistance of viscous secretions beneath the pedal sole) [4] and glue-like adhesion (via secretions that form chemical interactions to link the pedal sole with the surface) [11,14,15]. Following a series of studies ruling out suction in *Patella* [4,6,14], suction as a potential mechanism of attachment was re-introduced in a distantly related family of limpets, Lottidae [5,12]. The authors found a relationship between tenacity and ambient pressure, and also measured pressures directly beneath the pedal sole. Consequently, Smith proposed that limpets alternate between actively creating suction for adhesive locomotion at high tide and glue-like adhesion using adhesive mucus for powerful long-term attachment at low tide [5,11,12]. Such relationships have not been investigated in *Patella*. Furthermore, as Patellidae separated from Lottidae around 191 million years ago [1], it is unclear if this dual mechanism of suction and glue-like mucus is also used by *P. vulgata*.

If limpets do secrete a specialized pedal mucus for adhesion, a detailed biochemical characterization could offer insights into how the molecular components interact to function as a bio-adhesive. Limpet pedal mucus, independent of the species, is largely composed of water, around 90–95% wet weight, with the rest being proteins, carbohydrates and inorganic material [14–16]. Pedal mucus is highly resistant to solubilization [14,15], although addition of protein and

sugar-degrading enzymes results in a complete breakdown of the mucus [14]. Two different types of pedal mucus were identified from *Lottia limatula*: a solid plaque ('adhesive mucus') that was inconsistently left attached to the substrate when stationary limpets were forcibly detached, and more viscous pedal mucus from active limpets ('non-adhesive mucus') [11,15]. The biochemistry of the adhesive and non-adhesive mucus differed in two ways: first, there was around a twofold increase in protein and carbohydrate content from non-adhesive to adhesive pedal mucus [15]. Second, of the nine protein bands isolated from *L. limatula* pedal mucus, one protein (118 kDa) was present only in the adhesive samples, while a 68 kDa protein was associated with the non-adhesive mucus. The study concluded that *L. limatula* can control the properties of the mucus and transition from a non-adhesive to an adhesive type by modulating both the level and type of proteins secreted.

Although studies have examined the physical and biochemical properties of *P. vulgata* pedal mucus [14,17], none identified the different types of mucus as seen in *L. limatula*. Nine glands have been characterized from *P. vulgata* pedal sole, five of which can secrete pedal mucus into the contact zone (space between the pedal sole and the attachment surface) [17]. Histochemical tests indicated that proteins and sugars are stored within these glands, possibly as glycoproteins in some of them [14]. Although putative locomotory or adhesive functions were assigned to the glands, these designations were not experimentally validated. Eight proteins ranging from 23 to 195 kDa were extracted from one type of *P. vulgata* pedal mucus that is probably similar to 'non-adhesive' mucus from *L. limatula* based on sampling method [14]. This pedal mucus is a viscoelastic material, exhibiting fluid and solid-like behaviour, and is probably a cross-linked gel [14,18]. It is not soluble in water and requires strong reducing agents or harsh alkaline conditions for solubilization, and proteolytic or glycosidic enzymes for full degradation [14].

While these earlier efforts offer initial biochemical descriptions of the limpet pedal mucus, our knowledge of its molecular components and their function remains limited compared with our understanding of other marine bio-adhesive secretions. Advances in sequencing technology and bioinformatics have allowed researchers to assemble and analyse transcriptomes and proteomes in order to characterize the molecules and their interactions that govern bio-adhesive systems. Consequently, our understanding of marine bio-adhesives has drastically expanded over the last three decades (example publications, among many others, include [19–22]). However, the bulk of our knowledge stems from biological systems that use either temporary adhesion (e.g. sea stars, sea urchins, barnacle larvae and flatworms) or permanent adhesion (e.g. mussels, adult barnacles and macro-algae) [21]. Tidal transitory adhesion, as seen in limpets, requires different functionalities; limpets need to attach weakly during locomotion, but also form strong attachments for long periods of time. Another crucial difference that makes limpet adhesion special and worth investigating is that the long-term attachment is reversible or degradable for locomotion. At the same time, limpet adhesive mucus needs to withstand unwanted microbial degradation during stationary periods and potentially lower the risk of infection. A detailed molecular characterization of limpet pedal mucus, therefore, can help us understand how strong resistant biomaterials are synthesized and function as bio-adhesives.

In this study, we used a range of appropriate molecular biology approaches to investigate tidal transitory adhesion in *Patella vulgata*, including transcriptomics, proteomics, lectin-based assays and *in situ* hybridization. We isolated and identified 171 candidate adhesive protein sequences from different types of pedal mucus. We also localized specific sugar residues to pedal glands and secretions. Fourteen candidate protein sequences were individually annotated with conserved protein domains, many of which are also present in published temporary adhesives from marine invertebrates. To elucidate the role of active suction in *P. vulgata* and to complement our molecular investigation, we performed sub-pedal pressure recordings while limpets were freely locomoting, under stimulated predatory attack and during manual detachment via normal pull-offs. We conclude that *P. vulgata* limpets secrete pedal mucus that has similar molecular constituents to temporary bio-adhesives but with several clear distinctions. Furthermore, we found no evidence of large reductions in sub-pedal pressure, suggesting that suction is not the principal mechanism underlying the limpet's powerful attachment.

# 2. Material and methods

## 2.1. Animal collection and maintenance

Limpets (*Patella vulgata*) were collected on five occasions from Sheringham, England. Individuals with shell widths of around 20–35 mm were removed from exposed rock surfaces during low tide using the following approach to minimize damage to the pedal sole: a flat tool (e.g. flathead screwdriver) was placed at a shallow angle to the horizontal and slowly chiselled into the shell margin until the limpet was cleanly dislodged. The pedal soles of all individuals were visually inspected for damage before being brought to the laboratory. Limpets were kept in a marine aquarium tank lined with polyvinyl chloride acetate (PVCA) sheets for easy detachments, and a pump was scheduled on a timer to simulate tidal waves. Fit individuals (based on their resistance to dislodgement and clamping response) were detached without damaging their pedal soles for all the experiments.

## 2.2. *In vivo* pressure measurements

Pressure recordings were conducted *in vivo* to investigate the range of pressure differences generated beneath limpet pedal soles. Healthy individuals were placed on a custom-built underwater set-up consisting of a horizontal platform made from clear acrylic with four adjustable walls to constrain the path of the moving limpet (see electronic supplementary material, figure S1). A small circular hole (0.5 mm radius) was laser-cut in the horizontal acrylic platform and connected to a pressure sensor (PX26-30DV, Omega Engineering, UK; see electronic supplementary material for details on calibration) via flexible tubing. For recording pressures beneath the pedal sole during locomotion, a limpet was first detached from the main tank, its pedal sole gently cleaned with laboratory tissue to remove existing mucus, then placed on the platform so that it had to locomote anteriorly and over the pressure sensor. In a second set of experiments, a predatory attack was simulated using a method adapted from [8] to elicit a clamping response: a stainless steel ball bearing (10 mm radius; 32.3 g weight) was accelerated down a 6 cm path at

45° incline by gravity to fall on the shell of the limpet. Lastly, limpets were allowed to locomote over the pressure sensor, tapped on the shell to elicit clamping, then manually detached with a vertical pull. The voltage output from the pressure transducer was recorded using a USB DAQ (NI-6001, National Instruments, USA) and a MATLAB script (MATLAB version R2017b, The MathWorks Inc., MA, USA). Analysis of the recordings was conducted in MATLAB.

## 2.3. Adhesive mucus sampling for biochemical characterization

Previous studies used different techniques to sample pedal mucus: Grenon & Walker detached *P. vulgata*, wiped the sole clean and left them upturned for 30 min before collecting the secreted mucus [17]. The samples were pooled from multiple individuals for biochemical characterization. Smith *et al.*, on the other hand, differentiated between adhesive ('A') and non-adhesive ('NA') mucus: adhesive mucus was sampled by detaching *L. limatula* that had settled onto glass walls of an aquarium tank, whereupon roughly a third of the limpets left behind a solid 'glue' that remained firmly adhered on the glass, while non-adhesive mucus was sampled by placing several individuals in a plastic bag and making them move around without attaching firmly for 4–8 h [15]. Mucus samples from multiple *L. limatula* were then pooled for biochemical characterization.

For our experiments, we developed a specialized sampling technique to collect three different types of *P. vulgata* pedal mucus while minimizing contamination and damage to the individual (figure 2). Limpets were allowed to settle onto thin sheets of PVCA (around 200 μm thick) in an aquarium tank with circulating artificial saltwater (ASW, made per manufacturer instructions; Instant Ocean, Aquarium Systems, VA, USA) at the Biology of Marine Organisms and Biomimetics Unit, University of Mons, Belgium. Immediately before sampling, the plastic sheets with limpets were removed from the tank, cleaned of excess debris, and positions of each settled limpet were marked using a permanent marker. The limpets were then carefully detached from the thin PVCA sheet by peeling (figure 2*a*)—this minimized both damage to the soft pedal sole and possible contamination from other mucus types. Since our collection method was similar to that used to sample barnacle cement ('primary cement' is secreted naturally for attachment, while 'secondary cement' is secreted only during re-attachment [23]), we have adapted their terminology to define our three types of pedal mucus. The thin mucus layer found on the PVCA sheet, similar to the 'adhesive' mucus from Smith, was defined as 'interfacial primary adhesive mucus' (IPAM; figure 2*b*), while the mucus layer remaining on the pedal sole was termed 'bulk primary adhesive mucus' (BPAM). IPAM was collected by scraping it off the plastic sheets using sterilized glass squares with different extraction buffers (see below for buffer descriptions). BPAM was collected directly from the pedal soles of upturned limpets by making the thin mucus film swell with a small volume of filtered ASW (Instant Ocean, per manufacturer instructions), then removing the swollen mucus using sterilized forceps. The third mucus type was isolated using a previously described method [14] with some modifications: after the BPAM was collected, the pedal sole was thoroughly cleaned with ASW and laboratory tissue paper, then the limpet was left upturned in a humid container to stimulate fresh mucus production. After at least

royalsocietypublishing.org/journal/rsob  Open Biol. **10**: 200019

**Figure 2.** Three types of pedal mucus were sampled from individual *P. vulgata* limpets. (*a*) Limpets were allowed to settle onto thin PVCA films; (*b*) Interfacial primary adhesive mucus (IPAM) was collected by peeling a thin PVCA film from a settled limpet and scraping the thin layer on the PVCA, while bulk primary adhesive mucus (BPAM) was sampled from the limpet pedal sole; (*c*) Secondary adhesive mucus (SAM) was sampled by gently detaching a limpet, wiping the pedal sole clean, leaving it upturned for at least 30 min, then gently collecting the SAM from the pedal sole.

30 min, a small volume of newly secreted mucus, called 'secondary adhesive mucus' (SAM), was collected (figure 2*c*). It should be noted that as SAM was collected in air and its composition may vary if sampled from individuals left in an aqueous environment. Samples from individual limpets were kept separate and kept frozen at −20°C until further analysis.

## 2.4. Protein extraction and gel electrophoresis with various staining methods

The three different mucus types (IPAM, BPAM, SAM) were solubilized and extracted for gel electrophoresis using the protocol described previously with minor modifications [24]. The extraction buffer, made with 1.5 M Tris-HCl buffer at pH 7.8, 5 M urea, 2% (weight/volume) SDS, and 0.5 M dithiothreitol (DTT), was added to the samples and transferred to a glass pestle tissue homogenizer and manually ground for up to 5 min to thoroughly disrupt the mucus. The homogenized samples were then incubated at 60°C with agitation for 1 h, followed by 5 min of cooling down to room temperature (RT). Sulfhydryl groups were carbamidomethylated with iodoacetamide used in a 2.5-fold excess (w/w) to DTT in the dark at RT for 20 min. An equal quantity of β-mercaptoethanol was added to stop the reaction, and the sample was centrifuged at 13 000 RPM for 15 min at 4°C. The supernatant containing solubilized proteins was transferred to Eppendorf tubes and stored at −20°C for future use.

For gel electrophoresis, Laemmli sample buffer (Bio-Rad) was added to the protein extracts with 5% (v/v) β-mercaptoethanol, heated for 2 min at 90°C, then centrifuged for 5 min at 16 000 *g*. 10% sodium dodecyl sulfate (SDS)–polyacrylamide gels were used and subsequently stained with Coomassie Blue (to visualize proteins), Periodic Acid Schiff (PAS; to visualize carbohydrates) or Stains-All (colour depends on protein properties; blue for acidic proteins or $Ca^{2+}$-binding proteins [25], purple for proteoglycans, pink for less acidic proteins [26]).

## 2.5. Peptide sequencing using mass spectrometry

Mass spectrometry was performed as previously described [27] with minor modifications. Proteins from whole extracts of IPAM, BPAM and SAM from three individuals were precipitated using cold acetone (repeated until DTT scent was not detectable), then subjected to trypsin digest at 37°C overnight (1 μg of trypsin per 50 μg of extracted protein; modified porcine trypsin, sequencing grade from Promega). Tryptic peptides were analysed by reverse-phase HPLC-ESI-MS/MS using an Eksigent NanoLC 400 2D Ultra Plus HPLC system connected to a TripleTOF 6000 quadrupole time-of-flight mass spectrometer (AB Sciex, Concord, ON, Canada). After injection, peptide mixtures were transferred to a AB Sciex column (3C18-CL 75 μm × 15 cm) and eluted at a flow rate of 300 nl $min^{-1}$. MS data was acquired using the TripleTOF 6000 mass spectrometer fitted with a Nanospray III source (AB Sciex) using a pulled quartz tip as the emitter (New Objectives, MA, USA).

## 2.6. RNA isolation and transcriptome generation

Total RNA was isolated from an individual *P. vulgata* limpet by the following method: first, the pedal sole was carefully excised from surrounding tissue on ice and divided into longitudinal sections. Next, each section was immediately frozen in liquid nitrogen, added to TRIzol (Life Technologies, Carlsbad, CA), then homogenized using a hand-held mechanical tissue homogenizer. After going through the recommended protocol using TRIzol reagent, quality of the isolated RNA was initially checked using spectrophotometry, at which point a second clean-up was conducted using RNeasy Mini-kit (Qiagen, CA, USA) to digest genomic DNA and to further purify the RNA. Subsequent sequencing, data processing and transcriptome assembly were performed at the Beijing Genomic Institute, China (BGI). Integrity of the isolated RNA was assessed by gel electrophoresis and via Agilent Bioanalyser prior to sequencing (RNA integrity number: 6). Illumina HiSeqXTen platform was used to generate 150 bp paired-end reads, and the raw reads were filtered to remove adaptors and low-quality reads (see electronic supplementary material for more details). Cleaned reads were used for *de novo* assembly of the transcriptome with *Trinity* software v2.0.6 [28] and assembled into Unigenes with *Tgicl* v2.0.6 [29]. Fragments per kilobase of transcript per million mapped reads (FPKM) values were calculated by first mapping clean reads to Unigenes with

*bowtie2* [30] (v2.2.5, sensitive mode; see electronic supplementary material for full software settings), then *RSEM* [31] (v1.2.12, default parameters) was used to quantify expression levels. To assess transcriptome assembly and annotation completeness, we conducted an analysis based on the Benchmarking Universal Single-Copy Orthologs (BUSCO) using *BUSCO* v3.0.2 [32] for metazoa_odb9 and eukaryote_odb9 datasets. Based on the metazoan dataset, the assembled transcriptome was estimated to be 91.5% complete with 894 complete BUSCOs, 4.4% (43) fragmented BUSCOs and 4.1% (41) missing BUSCOs from a total of 978 BUSCO groups searched. Similar values were obtained with the eukaryota dataset. Note that these BUSCO numbers are in line with those from the *Lottia gigantea* reference genome [32]. Raw sequencing reads and the assembled transcriptome has been deposited to an NCBI BioProject database under accession number PRJNA613775.

## 2.7. Searching for peptides against the transcriptome

The assembled transcriptome was translated into the six open reading frames (ORF) on a Galaxy Project server (https://usegalaxy.org [33]). MS/MS data were searched for protein candidates against all ORFs using the software ProteinPilot 5.0 (AB Sciex). Carbamidomethyl cysteine was set as the fixed modification and trypsin as the digesting enzyme. For all samples, candidates with false discovery rates (FDR) above 0.01 were excluded from further analyses. Only sequences that appeared in all three individuals were compiled to form a single set of candidate proteins for IPAM, BPAM and SAM. Additionally, these three lists were combined, and duplicates removed to create a total candidate protein list obtained from aligned transcriptome and proteome data. Homology against known proteins was assessed using NCBI Basic Local Alignment Search Tool for protein (blastp) against UniProtKB/SwissProt databases (www.uniprot.org [34]) set to the default parameters. Conserved protein domains were searched on InterPro (v75.0) [35]. Clustal Omega (www.ebi. ac.uk/Tools/msa/clustalo) [36] and Jalview Version 2 (v2.10.4b1) [37] were used for multiple sequence alignments to check for conserved protein domains. The following suite of prediction algorithms were used to characterize the candidate proteins: DeepLoc 1.0 was used for localization predictions[38]; NetCGlyc 1.0, NetNGlyc and NetOGlyc 4.0 for C-, N- and O-glycosylation predictions [39–41]; NetPhos 3.1 for phosphorylation predictions [42]; and SulfoSite for predicting tyrosine sulfation sites [43].

## 2.8. Expression analysis of candidates using *in situ* hybridization

From the combined list of candidate proteins, a subset was selected for further analysis using *in situ* hybridization (ISH) based on the following criteria: first, we limited our selection to proteins that were ranked highly by ProteinPilot to ensure we were targeting proteins that were present in adhesive mucus. Second, we included candidates with conserved protein domains that were commonly associated with marine bio-adhesives (e.g. vWFD, EGF, lectins). Finally, we sought to sample proteins across the different types of mucus with the goal of identifying candidates associated with specific types of mucus (IPAM, BPAM and SAM). ISH probes were generated

based on a modified protocol from [44] and is described in full in the electronic supplementary material. In summary, cDNA was generated from isolated total RNA (same limpet individual as the one sent for transcriptome sequencing) using Transcriptor First Strand cDNA Synthesis Kit (Roche). Gene-specific primers were designed with Primer 3 (http://primer3.ut.ee, v4.1.0) and used to synthesize template DNA for the production of digoxigenin-labelled (DIG) antisense RNA probes (see electronic supplementary material for list of primers used). Limpet tissues were fixed in 4% paraformaldehyde (PFA) in PBS, embedded in paraffin wax, then sectioned into 14 µm sections using a Microm HM 340 E microtome. Probes were added to tissue sections and developed using NBT/BCIP system (Roche) at 37°C. Sections were mounted and imaged with a Zeiss Axio Scope.A1 microscope.

## 2.9. Lectin staining to identify specific sugar residues within limpet pedal sole

Lectin staining was used to provide additional insight into the identity and locality of specific sugar residues within the limpet adhesive organ (see [45] for additional information on lectin-based staining method to investigate adhesive organs). Nine biotinylated lectins (see table 1; Vector Laboratories, USA) were applied to 5 µm paraffin sections of *P. vulgata* pedal sole and visualized using Texas Red conjugated Streptavidin. For the negative control, a section was prepared alongside the rest, but no lectin stain was added. A few sections were also stained with alcian blue at pH 2.5 and counterstained with phloxine to facilitate the interpretation of the lectin stains by providing an overview of the pedal sole morphology and glands. Sample images were taken at the following regions to aid in comparison between different staining patterns: marginal groove, anterior (immediately posterior to the marginal groove), middle and posterior end of the foot. All images were taken with a Zeiss Axio Scope.A1 microscope. Qualitative assessment of lectin stain intensity was conducted based on images taken with the same exposure settings. Images for figures were post-processed in FIJI to enhance clarity [46].

# 3. Results

## 3.1. General observations on *Patella vulgata* attachment and *in vivo* pressure recordings

During the course of this study, we recorded four key observations about limpet attachments that provided novel insights into their adhesion: first, when stationary limpets were detached (by peeling from plastic sheets) and immediately placed onto a smooth flat surface (glass or plastic), their adhesion was insufficient to hold their own weight when turned upside-down (i.e. their re-attachment was not immediate). Once the limpets had time to locomote away from being returned to a surface (around 1–2 min), they consistently left behind a gel-like layer on the surface (probably similar to the BPAM collected for proteomics; see electronic supplementary material, video S1). Such samples resisted degradation in the saltwater tank for several weeks. Second, stationary limpets that were well attached remained so even when a length of wire was pushed through from one margin of the pedal sole to the other (electronic supplementary material, video S2).

**Table 1.** List of lectin-based stains used to investigate sugar residues present in the limpet pedal sole.

| lectin | acronym | target sugars | staining intensity | pedal sole | side-wall |
|---|---|---|---|---|---|
| *Lens culinaris* agglutinin | LCA | α-Man, α-Fuc linked to N-acetylchitobiose | +++ | glands | no |
| *Ulex europaeus* agglutinin I | UEA I | α-Fuc | + | epithelium, glands | no |
| wheat germ agglutinin | WGA | GlcNAc (dimers or trimers preferred), chitobiose Sialic acid | ++ | epithelium, glands (also throughout the foot) | epithelium |
| succinylated wheat germ agglutinin | sWGA | GlcNAc (without sialic acid) | +++ | glands | epithelium |
| *Ricinus communis* agglutinin I | RCA I | Gal, GalNAc | + | glands | no |
| *Maackia amurensis* lectin II | MAL II | (α-2,3)-sialic acid | + | glands | no |
| soy bean agglutinin | SBA | α- or β-GalNAc, Gal | + | non-glandular structures throughout the foot | no |
| Concanavalin A | ConA | α-Man | +++ | n.a.[a] | n.a. |
| Jacalin | Jacalin | Galβ-(1−>3)-GalNAc | — | no | no |
| none | negative control | n.a. | — | no | no |

[a]n.a.: not applicable due to unspecific staining; —: no staining observed.

Third, limpets crawled up the vertical wall of a basket made up of plastic fishing mesh placed in the aquarium and settled slightly above the waterline with strong adhesion (electronic supplementary material, figure S2). Lastly, limpets that died while firmly adhered to plastic sheets remained well attached, and the surface could be peeled away from them in a similar way to living limpets. Each of these observations suggested a mechanism of attachment that is not reliant on muscle-actuated suction, which is typically dynamic (fast attachments and detachments) [47], detaches from a disruption of the rim, fails to seal on meshes or porous substrates and is often not functional upon death of the animal.

To further investigate the contribution of reduced pressures beneath limpet pedal soles, sub-pedal pressures were measured while limpets attached to a clean smooth acrylic surface. Plots from a representative experiment with three different conditions (free locomotion, simulated attack and normal pull-off) are shown in figure 3. When limpets were freely locomoting, we recorded both positive and negative pressures (relative to ambient pressure) beneath the pedal sole, ranging from −0.79 to 1.0 kPa ($n = 20$ measurements from 8 limpets). The average (± standard deviation, s.d.) minimum and maximum recorded pressures per limpet were −0.42 ± 0.23 kPa and 0.32 ± 0.33 kPa, respectively ($n = 8$). No discernible difference in pressures was observed when limpets were left to attach to the surface for 10 min or longer. When limpets were disturbed with a simulated predatory attack, we saw a distinct reduction in pressure (−2.3 ± 1.5 kPa, average ± s.d.; $n = 6$ limpets). These pressure reductions decayed exponentially over time, taking 3.7 ± 3.0 s (average ± s.d.; $n = 8$ measurements from 5 limpets) to reach 60% of the minimum pressure. The most negative pressure value (−5.7 kPa) was recorded when an attached limpet was manually pulled off perpendicularly from the surface (see electronic supplementary material for additional information). The average pressure value for all manual detachments was −1.5 ± 1.9 kPa (± s.d., $n = 4$ limpets).

## 3.2. Overview of the molecular components of *Patella vulgata* pedal mucus

As a result of our revised mucus collection method (figure 2), we successfully isolated three types of limpet pedal mucus (IPAM, BPAM and SAM). We observed a number of qualitative differences between the types of mucus: first, IPAM was a thin layer left on the surface when the limpet was detached that sometimes felt like a raised solid patch. The thin layer of IPAM became visible with crystal violet staining. BPAM, on the other hand, was visible as an opaque swollen layer on top of the pedal sole and could at times be removed as an intact sheet of mucus. Lastly, the small quantities of SAM produced on the pedal sole easily broke apart during collection and did not form sheets like BPAM.

From the three types of pedal mucus collected from *P. vulgata*, we used gel electrophoresis to visualize multiple protein bands (electronic supplementary material, figure S3). In total, at least 11 distinct protein bands were identified with Coomassie Blue staining, with molecular weight estimates ranging from 40 to 190 kDa, and a few protein bands larger than 250 kDa. Unlike previous studies where mucus samples were pooled from many individual limpets, sufficient amounts of protein were collected to compare secreted proteins between individuals, although no discernible differences

royalsocietypublishing.org/journal/rsob    Open Biol. **10**: 200019

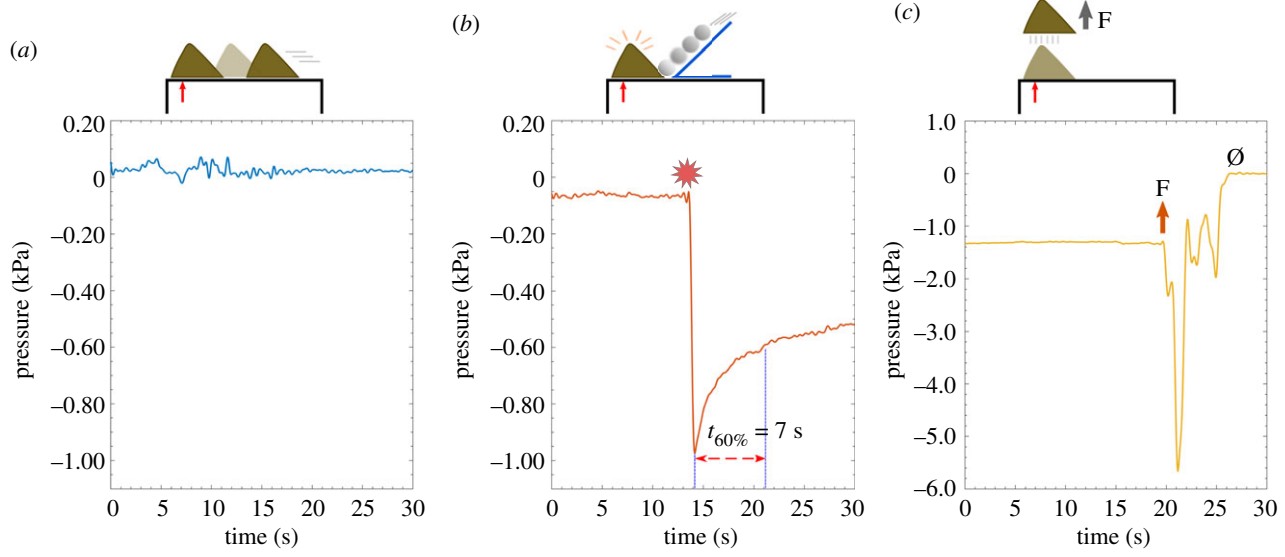

**Figure 3.** Representative *in vivo* sub-pedal pressure values from *P. vulgata* with schematics showing the different trial conditions. (*a*) Free locomotion: small pressure values (around −0.02 to +0.07 kPa) were observed when the limpet was undisturbed and locomoting over the sensor. (*b*) Simulated attack: when a ball bearing was used to simulate a predation event, the pressure was lower, at around −1.0 kPa. This negative peak decayed slowly, taking around 7 s to reach approximately 60% of the minimum pressure. (*c*) Normal pull off: when the limpet was allowed to settle over the sensor and then manually detached perpendicularly (arrow marks beginning of detachment), a sharp negative peak was recorded that reached −5.7 kPa, which returned to zero when the limpet detached (marked Ø). All tests were conducted under water.

were observed. The most prominent protein bands across all three adhesive mucus types were 45, 55, 130 and 160 kDa, and a protein band outside the top range of the ladder (referred to as 'greater than 250 kDa'). While proteins from the three mucus types were similar in size, one smeared band around 60 kDa was associated with the IPAM samples (electronic supplementary material, figure S3a). Periodic acid–Schiff (PAS) staining confirmed the presence of glycosylated proteins of approximately 60 kDa in size (electronic supplementary material, figure S3b). PAS staining also revealed large glycosylated protein-based complexes that were not fully disrupted and failed to migrate into the gel. Stains-all cationic dye provided further insights into the differences between the types of mucus (electronic supplementary material, figure S3c). We observed blue-stained protein bands (indicative of proteins that are highly acidic, negatively charged, and/or bind $Ca^{2+}$) at around approximately 110 kDa in both BPAM and IPAM, although the band stained more strongly in the latter. There was also a strong blue staining at the top of the gel that overlapped with the positive PAS staining, supporting the presence of large protein–sugar complexes that failed to enter the gel. Meanwhile, purple-stained bands (proteoglycans) at approximately 55 kDa and greater than 250 kDa and a bright pink band (weakly acidic proteins) of approximately 160 kDa were associated with BPAM only. Once again, there was a smeared band approximately 60 kDa from all IPAM samples, and this band stained pink to purple.

Lectin assays provided additional information on the nature of the sugar residues present in limpet pedal mucus (figure 4). Six of the nine tested lectins labelled specific glands and secreted mucus; four specifically targeted glands near the pedal sole and not the side-walls (table 1). We discovered that *Lens culinaris* agglutinin (LCA), which recognizes α-Mannose and/or α-Fucose linked to *N*-acetylchitobiose sugar residues, specifically stained glands within the pedal sole and not the side-wall. LCA revealed an anterior–posterior gradient of stained glands, where the anterior of the foot

featured the highest density of staining, followed by the posterior end of the foot, while the middle section was stained least intensely (figure 4e). We observed oval glands within the pedal sole epithelium, while flask-shaped subepithelial glands were found in a zone of high glandular density extending from the epithelium to approximately 150 μm into the body (dorsally). In contrast to the strong and dense staining from LCA, the rest of the lectin stains localized to more specific glands within the foot, the pedal sole or the side-wall. *Maackia amurensis* lectin II (MAL II) stains highlighted distinct granular contents within pedal sole glands, and succinylated wheat germ agglutinin (sWGA) stained granular gland contents distributed throughout the foot tissue as well as in pedal sole glands. Wheat germ agglutinin (WGA), unlike sWGA, did not stain pedal sole glands. Both WGA and sWGA strongly stained the epithelium. Notably, sWGA and LCA captured several glands in the midst of secreting mucus to the outside. We observed a dorsoventral gradient of glands with different sugar residues: while LCA, sWGA and *Ulex europaeus* agglutinin 1 (UEA I) stained glands both close to the epithelium and deeper (dorsally) into the tissue, *Ricinus communis* agglutinin I (RCA I) and MAL II localized to specific glands further away from the epithelium. MAL II-stained glands particularly deep in the limpet foot (approx. 60 to 150 μm away from the epithelium), with granular contents secreted through long necks to the outside.

## 3.3. Identification of putative limpet adhesive proteins from transcriptomics and proteomics

A *de novo* transcriptome was obtained from the *Patella vulgata* pedal sole (electronic supplementary material, table S1). Functional annotation was conducted using seven databases (NR, NT, GO, KOG, KEGG, SwissProt and InterPro), which yielded 37 261 annotations overall (see electronic supplementary material, table S2 for numbers from each database). This

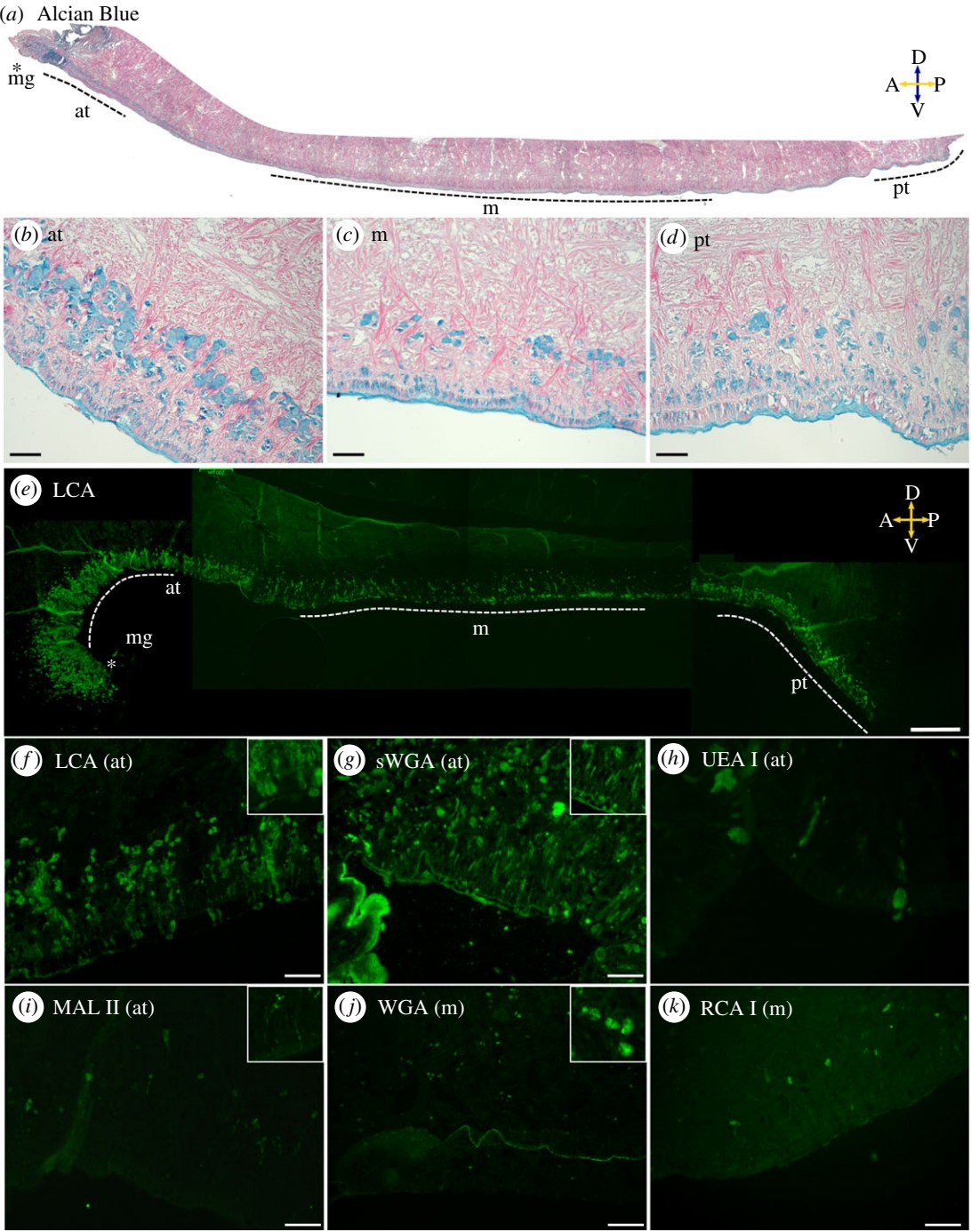

**Figure 4.** Overview of *P. vulgata* foot tissue and the chemistry of glandular secretions. (*a–d*) *Alcian blue* (pH 2.5) highlights glands in blue (carboxylate and sulfate moieties) and phloxine stains muscles in red. Regions used for higher-magnification images (*b–d*) are labelled as marginal groove (mg), anterior (at), middle (m) and posterior (pt). Scale bars: 100 μm in (*b–d*). Compass labels: A, anterior; P, posterior; D, dorsal; V, ventral. (*e–k*) Lectin stains highlight the different sugar residues present within specific glands. (*e*) LCA: stitched image showing the entire foot. Glands contain α-Man and/or α-Fuc linked to *N*-acetylchitobiose sugar residues. Side-wall glands are not stained. Dotted line marks the epithelium, and imaged regions are labelled as in (*a*). Scale bar 400 μm. (*f*) LCA (at): stained glands are found within the epithelium and up to approximately 300 μm into the foot. Scale bar 100 μm. Stained mucus and gland secretions are visible (inset, 150 μm box). (*g*) sWGA (at): secretions containing GlcNAc but not sialic acid are highlighted throughout the tissue. Scale bar 100 μm. Granules are secreted from long necks (inset, 200 μm box). (*h*) UEA I (at): specific glands contain α-Fuc. Scale bar 20 μm. (*i*) MAL II (at): infrequent glands deep in the tissue (approx. 100–150 μm) contain (α-2,3)-sialic acid. Scale bar 100 μm. Granules are secreted from long necks (inset, 370 μm box). (*j*) WGA (m): glands with GlcNAc are present throughout the foot tissue. Note the epithelium is strongly stained. Glands approximately 30 μm in size are full of granules (inset, 150 μm box). (*k*) RCA I (m): similar to MAL II but with GalNAc or galactose. Scale bar 50 μm. See text and table 1 for additional information on the lectins used and their ligands.

transcriptome was used as a reference to map peptide sequences of IPAM, BPAM and SAM protein extracts from pedal mucus (*n* = 3 individuals). With this approach, 171 candidate sequences for limpet adhesive proteins were identified: 27% of them (46 sequences) were found in all three mucus types (to classify as being present in a type of mucus, a candidate sequence had to be found in all three

individuals), while 22% (37 sequences) were present only in BPAM, 13% (23 sequences) in SAM and 9% (15 sequences) only in IPAM. Note that, due to our selection criteria (where a candidate protein had to be present in all three limpet individuals in order to be attributed to BPAM, SAM or IPAM), some proteins may not have been assigned to a particular type of adhesive mucus. Nevertheless, this

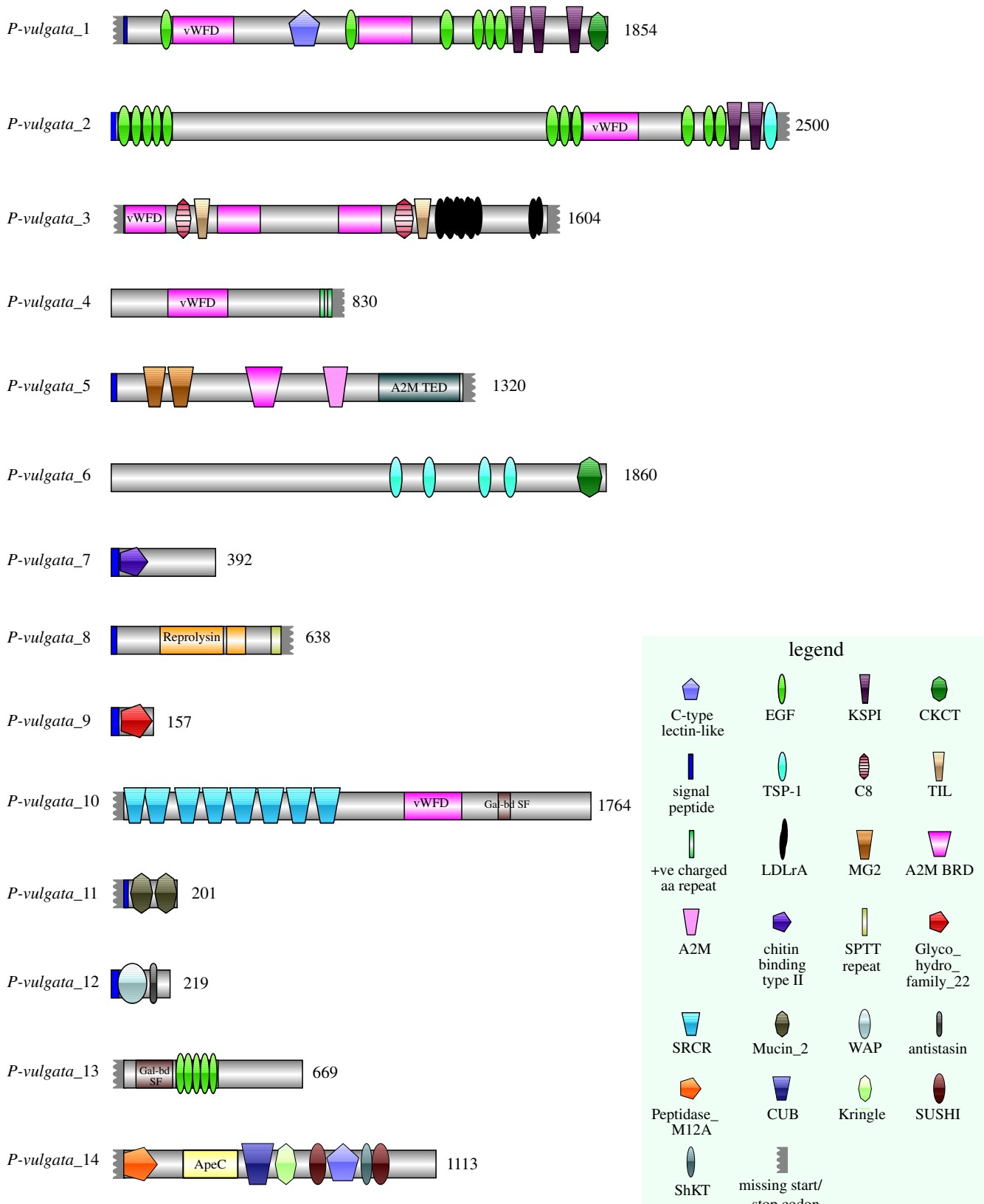

**Figure 5.** Conserved protein domains present in a subset of limpet pedal mucus proteins. Amino acid lengths are shown after the N-terminal of each sequence. See legend for details.

categorization helped to form initial ideas about the potential functions of each isolated protein.

Fourteen sequences (*P-vulgata_1* to *14*; see figure 5) were chosen for a more detailed analysis with manual annotation of conserved protein domains and *in situ* hybridization. Six of the 14 sequences were found in all three pedal mucus types (*P-vulgata_1* to *6*); one was found in IPAM and BPAM

only (*P-vulgata_7*); one from IPAM and SAM (*P-vulgata_11*); four from only IPAM (*P-vulgata_8* to *10*, *P-vulgata_12*), and one from each of the SAM and BPAM samples (*P-vulgata_13* and *P-vulgata_14*, respectively).

The following characteristics of the 14 sequences are summarized in table 2: the type of mucus it was isolated from (IPAM, BPAM, SAM), fragment per kilobase million (FPKM),

**Table 2.** Summary of putative adhesive proteins and their characteristics. See figure 4 and text for details.

| sequence ID | found in which mucus types? | FPKM | length (amino acids) | Start/ Stop codon | cysteine content (% of length) | signal peptide | predicted localization | conserved protein domains (InterPro) | homologous proteins of interest (BLASTp) | predicted glycosylation | predicted phosphorylation | predicted sulfation |
|---|---|---|---|---|---|---|---|---|---|---|---|---|
| P-vulgata_1 | all | 737 | 1854 | N/Y | 7.6% | Y | secreted | vWFD x2, C-type lectin-like, EGF x6, cystine-knot C-terminal, KSPI x3 | fibrillin, zonadhesin, alpha-tectorin, IgGFc-binding protein | N, C, O | numerous | 1 |
| P-vulgata_2 | all | 28 | 2500 | Y/Y | 7.8% | Y | cell membrane | vWFD, EGF x11, KSPI x2, TSP1 x1 | fibrillin, zonadhesin, alpha-tectorin, IgGFc-binding protein, **adhesive protein 1 [Minona ileanae]** | N, C, O | numerous | 5 |
| P-vulgata_3 | all | 2[a] | 1604 | N/Y | 8.6% | N | cell membrane | C8 domain x2, vWFD x3, TIL x2, LDL receptor class A binding repeats x5 | SCO-spondin, hemolectin/hemocytin, mucin | N, O | numerous | 0 |
| P-vulgata_4 | all | 17 | 830 | Y/Y | 5.7% | N | secreted | vWFD | zonadhesin, alpha-tectorin, IgGFc-binding protein, **Sfp-1 [Asterias rubens]** | 0 | numerous | 0 |
| P-vulgata_5 | all | 334 | 1320 | Y/N | 0.6% | Y | secreted | M2-like, A2M BRD, A2M, A2M TED | CD109-antigen-like, **SIPC [Megabalanus coccopoma]** | N, O | numerous | 0 |
| P-vulgata_6 | all | 1147 | 1860 | Y/Y | 5.9% | N | cell membrane | TSP1 x4, cystine-knot C-terminal | SCO-spondin, hemicentin, **adhesion protein 2 [Macrostomum lignano]** | N, C, O | numerous | 2 |
| P-vulgata_7 | IPAM & BPAM | 21 122 | 392 | Y/Y | 5.9% | Y | secreted | chitin binding type II | BSMP14 | N, O | numerous | 0 |
| P-vulgata_8 | IPAM | 180 | 638 | Y/N | 3.9% | Y | secreted | reprolysin | ADAM family mig-17 | N, O | numerous | 0 |
| P-vulgata_9 | IPAM | 2607 | 157 | Y/Y | 5.7% | Y | secreted | Glyco_hydro family 22 | C-type lysozyme | N, O | numerous | 0 |
| P-vulgata_10 | IPAM | 9 | 1764 | N/Y | 5.2% | N | cell membrane | SRCR x8, vWFD, galactose-binding domain superfamily | deleted in malignant brain tumours-1, scavenger receptor cysteine-rich protein type 12 precursor | N, C, O | numerous | 4 |
| P-vulgata_11 | IPAM & SAM | 4 | 201 | N/Y | 6.0% | Y | secreted | Mucin x2 | oikosin-like, mucin, cartilage intermediate layer-like | 0 | numerous | 1 |
| P-vulgata_12 | IPAM | 399 | 219 | Y/Y | 15.1% | Y | secreted | WAP, antistasin-like | n.a. | N, O | numerous | 0 |
| P-vulgata_13 | SAM | 162 | 669 | N | 9.7% | N | secreted | galactose-binding domain superfamily, EGF x5 | n.a. | N, O | numerous | 4 |
| P-vulgata_14 | BPAM | 65 | 1113 | N/Y | 4.9% | N | secreted | peptidase_M12A, ApeC, CUB, Kringle, SUSHI x2, C-type lectin-like, ShKT | n.a. | N, C, O | numerous | 1 |

[a]The sum of the FPKMs of two nearly identical contigs for P-vulgata_3 are shown.

**Figure 6.** Comparison between *P-vulgata_3* and SCO-spondin from *Gallus gallus*. *P-vulgata_3* shares many conserved domains with the first one-third of SCO-spondin sequence.

sequence length in amino acids, presence of start and stop codons, cysteine content, signal peptide, predicted subcellular localization, homologous proteins of interest based on NCBI blastp, conserved protein domains based on InterPro, predicted glycosylation (N-, C- or O-linked), predicted phosphorylation, and predicted sulfation. The cysteine content ranged from 3.9% to 15.1%, with the exception of one protein that had a low content of 0.6% (*P-vulgata_5*). The most common protein domains included von Willebrand factor type D (vWFD), epidermal growth factors (EGF), Kazal-type serine protease inhibitors (KSPI) and scavenger receptor cysteine-rich (SRCR). All proteins were predicted to have post-translational modifications (PTMs), with at least one type of glycosylation and numerous phosphorylation sites. While half of the proteins were predicted to have sulfated tyrosine, this PTM was not detected in *P. vulgata* (neither on foot sections nor on mucus trails) using an anti-sulfotyrosine antibody [48].

BLAST analysis of the limpet proteins highlighted homology to a number of characterized proteins. Most notably, three proteins (*P-vulgata_1, 2* and *4*) had the same group of homologous proteins: fibrillin, zonadhesin, alpha-tectorin and fragment crystallizable region of immunoglobulin G (IgGFc)-binding protein. These proteins can participate in ligand-binding, adhesion, oligomerization or fibril formation: for example, zonadhesin is a multi-domain protein believed to facilitate the binding of sperm to the egg zona pellucida [49], while fibrillin is a large $Ca^{2+}$-dependent glycoprotein that forms microfibrils in the extracellular matrix [50]. Moreover, both *P-vulgata_2* and *4* had sequence similarities to known adhesive proteins: adhesive protein 1 from *Minona ileanae* (QEP99777.1) [51] and sea star footprint protein 1 (Sfp-1) from *Asterias rubens* (AHN92641.1) [24], respectively. It is worth mentioning that *P-vulgata_4* featured two positively charged repeats at the C-terminus (amino acid sequence RRSRRNRNKARRSRRNRN) that did not align with any of the homologous proteins.

Two proteins—*P-vulgata_3* and *6*—were similar to SCO-spondin, a large secreted glycoprotein from the thrombospondin family involved in neural development with binding sites for sugars, proteins and lipids [52,53]. *P-vulgata_3* in particular was highly homologous to SCO-spondin, with homology to NCBI reference sequences for SCO-spondin from multiple unrelated species (e.g. 84% QC and 30.70% ID to SCO-spondin from *Gallus gallus*, NP_001006351.2; see electronic supplementary material, figure S4a). Interestingly, *P-vulgata_3* was highly similar to a specific portion of SCO-spondin (figure 6). The implications of this homology are discussed in the following section. While *P-vulgata_3* featured nearly all of the conserved domains of SCO-spondin (vWFD, TIL, C8, LDLrA), *P-vulgata_6* had just the repeating TSP-1-like domains in common. As a result of these repeating domains, however, *P-vulgata_6* aligned with a small portion of adhesion protein 2 from *Macrostomum lignano* (QAX24810.1) [54].

*P-vulgata_5* was homologous to the settlement-inducing protein complex (SIPC) from barnacles (*Megabalanus coccopoma*; 91% QC, 27.70% ID; BAM28692.1), both of which contain alpha-2 macroglobulin domains. This protein's homology to SIPC is discussed in the subsequent section.

*P-vulgata_8* featured one type of domain, reprolysin, which is a metallopeptidase (also called adamalysin M12B peptidase). Clustal alignment with reprolysin consensus sequence cd04267 from NCBI Conserved Domain Database [55] highlighted similarities between the proteins but, importantly, the conserved catalytic HEXXH motif was absent in in *P-vulgata_8* (electronic supplementary material, figure S4b). In addition, unlike other adamalysin peptidases, *P-vulgata_8* featured nine tandem Ser-Pro-Thr-Thr repeats starting from residue position 598. This was probably incomplete as the stop codon was missing and the sequence terminated after the first Serine of the next repeat (i.e. S/PTT). Enrichment of Ser, Pro and Thr residues began upstream of the SPTT repeat region (amino acid 550–638) and accounted for 73 of the last 89 residues (82%) of the sequence.

*P-vulgata_9* contained a predicted enzymatic domain, glycoside hydrolase family 22 (Glyco_hydro_family_22). This sequence was similar to a C-type lysozyme from *Haliotis discus hannai*, a species of abalone (92% QC, 48.99% ID; ADR70995.1), as well as with an NCBI reference sequence for lysozyme C from *Canis lupus familiaris* (88% QC, 38.03% ID; NP_001300804.1). The conserved catalytic Glu residue found in lysozymes was present in *P-vulgata_9*.

Due to the tandem SRCR repeats, *P-vulgata_10* aligned with an NCBI reference sequence for the deleted in malignant brain tumours-1 protein isoform c precursor from *Homo sapiens* (48% QC, 46.37% ID; NP_060049.2). Other notable homologues included SRCR-rich proteins from sea stars (*Asterias rubens*, QAA95957.1), sea urchins (*Strongylocentrotus purpuratus*, NP_999762.1) and sea anemones (*Exaiptasia pallida*, Aipgene2358 [56]).

No reliable homologues outside of hypothetical proteins were identified for *P-vulgata_12, 13* and *14*. However, it is worth noting that the ShKT domain prediction in *P-vulgata_14* was unexpected as it was first described as a potent ion channel toxin in sea anemones [57]. Multi-sequence Clustal alignment of *P-vulgata_14* with four roundworm mucin proteins homologous to ShkT (Tc-MUC-1 to 4) [58] confirmed the presence of the conserved cysteine residues (electronic supplementary material, figure S4c). However, the catalytic dyad Lys-25/Tyr-26 found in ShKT was absent in *P_vulgata-14* and Tc-MUC-1 to 4.

## 3.4. Expression of putative limpet adhesive proteins

From the list of 171 candidate sequences isolated from the three mucus types, 16 were selected for *in situ* hybridization to determine their expression site within the limpet. These 16 sequences were selected based on their relative ranking,

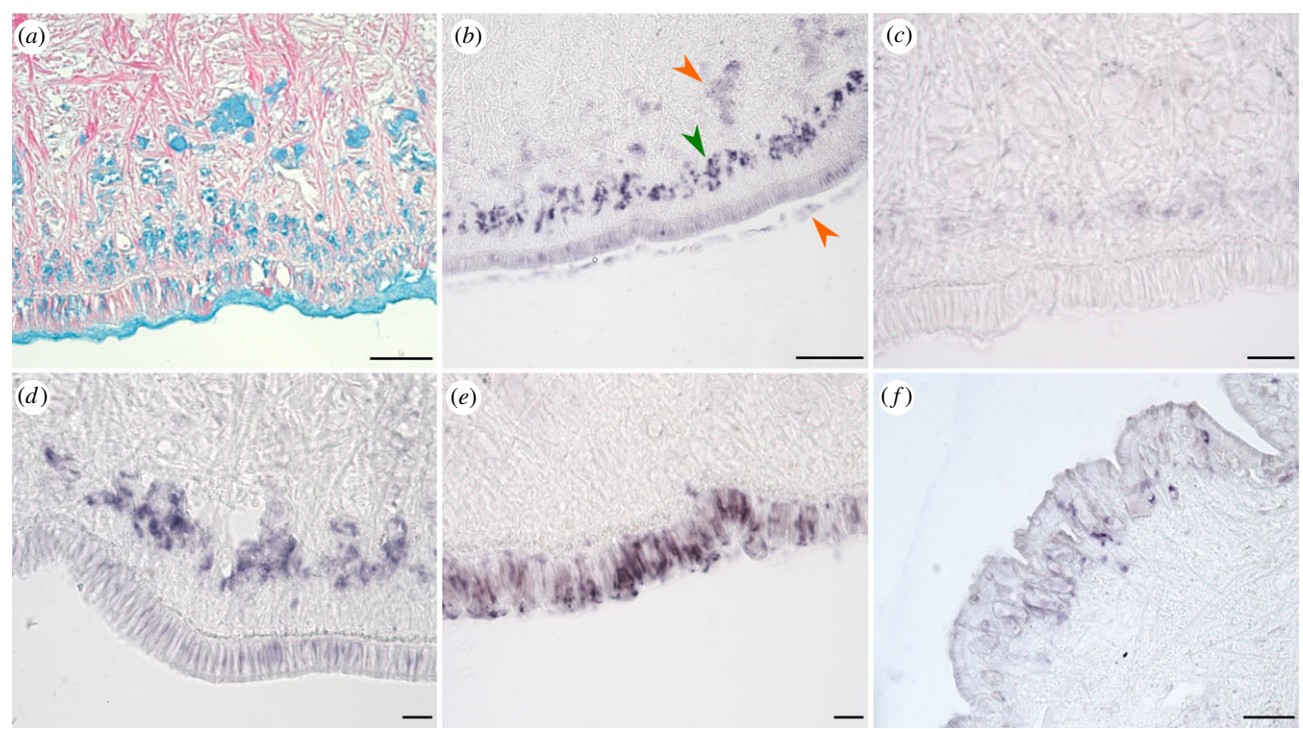

**Figure 7.** *In situ* hybridization (ISH) of five protein sequences confirms the presence and locality of the target mRNA within *P. vulgata* foot. (*a*) Alcian blue stain highlights the glands present in the foot and provides context for the ISH stains. Scale bar 100 μm. Probes for *P-vulgata_1* (*b*), *P-vulgata_3* (*c*) and *P-vulgata_4* (*d*) localized to a specific band of glands approximately 30–110 μm away from the epithelium. Scale bars 100 μm, 20 μm, 20 μm, respectively. Note that in (*b*), the weak staining around 150 μm away from the epithelium and in the mucus is unspecific background staining (orange arrowheads) that is distinct from the specific expression sites at approximately 100 μm away from the epithelium (green arrowhead). (*e*) *P-vulgata_7* stained the pedal sole epithelium. Scale bar 20 μm. (*f*) *P-vulgata_11* localized at the side-wall epithelium. Scale bar 50 μm.

domain similarity to published adhesive proteins and presence in a specific mucus type (electronic supplementary material, table S3). Specific expression patterns were obtained for *P-vulgata_1*, *P-vulgata_3*, *P-vulgata_4*, *P-vulgata_7* and *P-vulgata_11* (figure 7). As some of our samples had unspecific background staining, we only analysed those that produced distinct expression patterns (see figure 7*a*, which shows the distinct specific and unspecific regions). A more detailed account of the background staining is provided in electronic supplementary material, figure S5. From the five samples with distinct expression sites, *P-vulgata_1*, *P-vulgata_3* and *P-vulgata_4* stained a band of glands between approximately 30 μm and approximately 110 μm away from the epidermis, while *P-vulgata_7* localized to glands within the epidermis. These expression patterns confirm that the corresponding proteins were produced and secreted from glands specific to the pedal sole. *P-vulgata_11* specifically stained glands in the side-wall, which might indicate a contamination of the samples with side wall secretions.

## 4. Discussion

### 4.1. The role of sub-pedal pressure differences in limpet attachments

Numerous studies have sought to understand the principles behind the limpet's powerful attachment by ascribing it to suction [5,11,12], clamping [13] or glue-like secretions [11,14,15]. Smith reported a significant reduction in sub-pedal pressures when limpets (either *Tectura scutum* or *Lottia gigantea*) were placed on an acrylic surface and slid across the pressure gauge (approx. −20 kPa, relative to ambient), and an even

larger reduction when irritated to illicit a clamping response (close to −50 kPa) [5]. Our sub-pedal pressure differential measurements using *Patella vulgata* revealed much smaller values, with the average minimum pressure during free locomotion of −0.42 ± 0.23 kPa. Even with a simulated predatory attack inducing a clamping response, our values (−2.3 ± 1.5 kPa) were smaller than those reported by Smith. Our locomotion pressures are in good agreement with previous sub-pedal pressure measurements from locomoting *P. vulgata* (reported as 6 cm of water, which is around 0.6 kPa) [59]. When a predatory attack was simulated, the limpet clamped its shell against the surface, resulting in a rapid decrease in sub-pedal pressure, followed by a less negative pressure as the limpet relaxed slightly but did not quickly return to ambient (taking around 3.7 s to reach 60% of the peak minimum value). This response is similar to the clamping behaviour of *Cellana tramoserica* reported by Ellem *et al.*, where initial irritation (single tap to the limpet shell or the experimental set-up) caused a clamping force of 2–5 N that decayed in 1–2 s to pre-stressed levels, while continued irritation (continuous tapping) resulted in a much higher force of 25 N that decayed slowly over 5 min [13]. While we measured the sub-pedal pressures and not the forces, we observed a similar sustained decay in response to a simulated predatory attack. Hence, when our sub-pedal pressure findings are considered in combination with our observations on limpet attachment (i.e. adhering strongly with a disrupted seal or climbing on a mesh but not being able to re-attach quickly), it is likely that *P. vulgata* does not actively generate suction for attachment. Rather, it is likely that, at least in this limpet species, the most critical element for adhesion is creating a strong yet reversible connection between the pedal sole and the surface that is independent of sub-pedal pressures. Consequently, when the

limpet clamps down, the sole is able to sustain the reaction force even when a seal is absent. If a continuous seal is present or if influx is sufficiently low, then clamping will result in a pressure reduction that may further contribute to the overall attachment; however, our pressure recordings and behavioural observations indicate that low sub-pedal pressures are not required for attachments in *P. vulgata*.

## 4.2. Generating adhesion through chemical interactions: how limpet pedal mucus may function as a bio-adhesive

Several studies have previously investigated the chemical constituents of limpet pedal mucus using histochemistry and gel electrophoresis [14,15,17,48]. Our results from lectin-binding assays and transcriptome-assisted proteomics have revealed novel insights into the biochemical properties of *P. vulgata* pedal mucus.

Lectin staining of *P. vulgata* revealed a highly complex glandular system within the foot. We found LCA to be a good candidate for a comprehensive labelling of pedal sole glands. This suggests that LCA's target sugar residues (α-Man and α-Fuc linked to N-acetylchitobiose) are specific to the pedal sole mucus and not to secretions from other parts of the body. Furthermore, we observed a dorsoventral gradient of lectin staining, where the longest subepithelial glands (whose cell bodies are located approx. 60—150 μm dorsally from the pedal sole epithelium) are labelled with RCA I and MAL II (*N*-acetylgalactosamine or sialic acid, respectively). The density of these glands was also lower than that of the shorter ones expressing other residues, such as mannose, fucose, chitobiose, sialic acid or *N*-acetylglucosamine, which were found closer to the epithelium. At the other end of the spectrum, some glands were present within the epithelium, containing fucose, *N*-acetylglucosamine, chitobiose or sialic acid residues. A dorsoventral gradient may be important for the timing of secretions: glands close to the surface would require less time to secrete than those deep inside the foot, where the secretion needs to be pushed out through long thin necks. Spatial distribution of glands may also be important for the digital mucus glands of tree frogs, where clusters of ventral and dorsal glands within the attachment organ, the toe pad, have distinct morphologies and locations that may be related to their functions [60]. The tubular ventral glands, which supplies mucus to the surface of the adhesive toe pad, are situated deep within the organ and have long thin necks to the ventral surface, similar to the glands found in limpets. An alternative explanation for the dorsoventral gradient may be related to the volume of the glands: one way to increase the gland volume is to lengthen it with a long neck. Indeed, tubular glands are significantly larger than dorsal glands in tree frogs, and this is also likely to be the case for limpets. Our findings, which highlight the usefulness of lectins in clarifying limpet gland morphology and chemistry, also emphasize the need for future work on understanding the function of these glands and associated sugar chemistries.

From our transcriptome-assisted proteomics study of limpet pedal mucus, 171 sequences were identified from the limpet pedal mucus, of which 14 were selected and manually characterized. To facilitate discussion about their putative functions, these proteins have been assigned to three broad functional categories based on their domain composition:

(i) proteins likely to be involved in oligomerization, ligand binding (proteins, sugars, metals) and peptide stabilization (disulfide bridges); (ii) enzymes or inhibitors; and (iii) proteins with elements of both. It is worth mentioning that the transcriptome was based on the sequencing data from the pedal sole of a single limpet specimen. Since adhesive proteins are often large and repetitive [24,54], they tend to be inadequately assembled with short-read transcriptomics [51,61]. We sought to increase mapped transcript lengths by reducing the complexity of the input RNA and minimizing transcript variation caused by pooling samples from multiple individuals. Although our analysis showed that the transcriptome is of good quality, we want to highlight that due to the limited sample size, some transcripts and transcript variations may not be represented in this dataset.

Proteins *P-vulgata_4, 6, 7, 10, 11* and *13* fall into the first category, sharing among them domains associated with multi-protein complex formation (vWFD, EGF, SRCR and TSP-1) and protein–carbohydrate binding (C-type lectin-like, galactose binding-like domain superfamily and chitin-binding type II). The ability to bind to other proteins and carbohydrates (either free-existing or attached to glycosylated proteins) is an important requirement for the cohesion of gastropod mucus and underwater adhesives, which are often cross-linked and are difficult to dissolve without potent denaturing agents [14,62–66]. Furthermore, these proteins had elevated cysteine residue contents of 5–9.7%, much higher than the average for eukaryotes (1–2%), although this may decrease when the full length of some of the proteins are sequenced. Nevertheless, as all limpet mucus samples were challenging to dissolve even in harsh extraction buffers and high heat, disulfide bridges (either intra- or intermolecular bonds) are probably present in limpet adhesive mucus. Besides cohesion, ligand interactions are also important for adhesion to surfaces, and the carbohydrate-binding domains may promote interactions with surface-adsorbed or biofilm-based sugars [67].

Proteins *P-vulgata_5, 8, 9, 12* and *14* belong to the second category, with each sequence having at least one enzymatic (sugar-cleaving glycoside hydrolase and metallopeptidase) or inhibitory domain (WAP, antistasin, macroglobulin and ShkT). *P-vulgata_8*, while sharing some conserved residues with other members of the adamalysin metallopeptidases, lacks the canonical catalytic HEXXH motif. It is unclear, therefore, if *P-vulgata_8* has any enzymatic function. This protein warrants further investigation, however, as it was found only in IPAM. Similarly, *P-vulgata_9* was also found in IPAM and is a secreted protein with a glycoside hydrolase domain that cleaves sugar bonds within carbohydrates or linked to a glycoprotein. Unlike *P-vulgata_8*, this protein did include the catalytic residue, and may be involved in active degradation of pedal mucus to transition from stationary to locomotive states. Alternatively, since *P-vulgata_9* is homologous to lysozymes, it may have a defensive function, similar to cp-16 k, a lysozyme-like protein found in barnacle cement [68]. Follow-up studies on the activity of *P-vulgata_8* and *9* are necessary to understand their respective roles in limpet IPAM.

*P-vulgata_12* and *14*, on the other hand, are more likely to serve a defensive role. *P-vulgata_12* contains a WAP domain, which is often found in proteins with antiproteinase and antimicrobial activities [69,70], as well as an antistasin-like domain, a serine protease inhibitor (InterPro accession number IPR004094). Both domains contain multiple intramolecular disulfide bridges, which may afford stability in harsh

physical conditions [71]. *P-vulgata_12* therefore may be a highly stable protease inhibitor present in IPAM to protect the protein- and sugar-rich pedal mucus against foreign degradation. *P-vulgata_14*, on the other hand, is notable as it contains numerous domains for recognizing, binding and degrading sugars and peptides. One of its domains, Peptidase_M12A, is a zinc-dependent metallopeptidase, while Kringle, CUB and SUSHI domains are involved in recognition processes and regulating proteolytic functions. Apextrin-like (ApeC) and C-type lectin-like domains are both involved in innate immune responses of invertebrates by binding to bacterial peptidoglycans [72–74], although it should be noted that C-type lectin-like domains can interact with other types of ligands, such as proteins, lipids and inorganic compounds [74]. One unexpected domain prediction is the ShKT, which is a potent potassium ion channel inhibitor originally characterized from the sea anemone *Stichodactyla helianthus*. Loukas *et al.* found homologous regions (referred by its alternative name, six-cysteine repeat, SXC) from four proteins (Tc-MUC-1 to 4) encoding for secreted mucins in the parasitic nematode *Toxocara canis* with no toxin-like function [58]. Our analysis of *P-vulgata_14*, Tc-MUC-1 to 4, and ShKT confirmed sequence homology to the ShKT domain; however, like Tc-MUC-1 to 4, *P-vulgata_14* lacks the functional dyad found in the ShKT toxins and is unlikely to function as a potent ion channel inhibitor. Instead, the domain could be involved in forming stable disulfide bridges that occurs in the native structure of the ShKT. The abundant target recognition and regulation-related domains suggest that *P-vulgata_14* acts as an antibacterial agent within the pedal mucus. Interestingly, since this protein was found only in the BPAM samples, possible roles of *P-vulgata_14* are to: (i) prevent microbial degradation of the secreted mucus, which has to remain functional over prolonged periods of time (e.g. during high tide, when the limpet typically stops foraging and remains stationary within the safety of its home scar), and (ii) to minimize risk of infection. However, many aspects of this protein need to be further investigated to verify its purported function, such as its target specificity, stability and how it interacts with the gel network.

*P-vulgata_1*, 2 and 3 belong to the third category, with numerous domains for diverse ligand-binding in combination with protease inhibitor domains. These proteins represent some of the longest, most well-represented and complex sequences from the annotated set. Each protein contains at least four different domains and can potentially interact with proteins (through $Ca^{2+}$-binding EGF, TSP-1, vWFD), glycans (C-type lectin-like domain) or lipids (through lipoproteins binding to LDLrA). These domains suggest that these proteins can bind to diverse ligands that are soluble and/or adsorbed to the surface, as well as undergoing homo- or hetero-oligomerization (for example, vWFD, lectins and cystine knot, C-terminal domains can self-oligomerize [74–76]). Such functional domains could promote the formation of large networks with protein–protein and protein–glycan cross-links, which is essential for cohesive strength and is a mechanism to interact directly with surfaces or surface-adsorbed molecules [67]. This hypothesis is supported by the type of homologous proteins identified through blastp analysis: *P-vulgata_1*, 2, as well as 4, shares homology with fibrillin, zonadhesin, alpha-tectorin and IgGFc-binding protein. Incidentally, the similarities between these proteins have been reported previously in a study on the evolution of gel-forming mucin proteins [77]. These proteins form microfibrils (fibrillin;

[50,78]), bind to glycoproteins (zonadhesin [79]), recruit collagen fibrils to microvillar membrane surfaces (tectorin [80]) and form gels (IgGFc-binding protein [81]).

Similarly, *P-vulgata_3* is highly homologous with the consensus sequence for SCO-spondin, albeit only to roughly one-third of the total length. SCO-spondin is a large secreted glycoprotein present in the central nervous system and can be found either in a soluble state or aggregated into Reissner's fibre [53,76]. The homologous segment contains vWFD, C8, TIL and LDLrA domains, while the remainder features numerous copies of TSP-1, SCO-spondin region repeats (SCOR), TIL and a CTCK [76]. Interestingly, the TSP-1-rich segment from SCO-spondin that is absent in *P-vulgata_3* is specifically implicated in promoting neuronal development [82,83], which appears to be a superfluous function for adhesive proteins. Furthermore, *in situ* hybridization localized *P-vulgata_3* expression to the pedal sole, ruling out contamination as a possible source of SCO-spondin-like peptides. Hence, *P-vulgata_3* may represent a re-purposing of a highly conserved protein involved in neuronal development to an adhesive one through the loss of the TSP-1 and SCOR repeats. Alternatively, the TSP-1 motifs in SCO-spondin sequences have been increasingly duplicated through evolution, as evident in the lengthening observed in SCO-spondin-like proteins from Echinodermata to Vertebrata [84]. Ancestral SCO-spondin therefore may have served a conserved role in adhesion, then gradually lengthened to facilitate an increasingly important function in neuronal development. Indeed, the domains that were conserved, including vWFD, C8, TIL and LDLrA, are adhesive-like by themselves and can participate in protein–protein interactions; for example, both C8 and TIL domains can form interdimer disulfide bridges with vWFD [85].

Unlike the other annotated limpet adhesive proteins, *P-vulgata_3* is unique in having five highly conserved LDLrA repeats that can bind to multiple targets other than lipoproteins, including glycoproteins like TSP-1 and reelin [53]. Lipids are believed to be an important component of barnacle larvae and mussel permanent adhesives [86,87]. In barnacle larvae, lipidaceous granules (probably in the form of lipoproteins or lipopolysaccharides) are secreted as a primer to potentially displace water from the contact surface and to provide a protective and stabilizing environment for the subsequent proteinaceous secretion [86]. Similarly, limpet pedal mucus probably contains lipidic moieties (V. Kang 2020, unpublished data), which suggests an intriguing parallel between permanent and tidal transitory adhesives that has yet to be verified in temporary adhesive systems (but see [54,88]). However, more work is needed to understand the role of *P-vulgata_3*, including its relative abundance within the different types of adhesive mucus. Although its transcript expression level was low (table 2), it is difficult to draw conclusions about protein abundance based solely on the transcript expression levels, especially when the FPKM is derived from a single individual. Follow-up studies using techniques like exponentially modified protein abundance index (emPAI) can provide quantitative information on the abundance of *P-vulgata_3* and other adhesive proteins.

## 4.3. Comparing adhesive proteins from marine invertebrates

Limpets exhibit tidal transitory adhesion, where they transition from high-strength, semi-sessile attachment to locomotory

attachment. While it is currently not feasible to tease apart all the nuances of what makes a protein suitable for temporary, transitory or permanent adhesion, our results offer some useful initial insights: (i) both transitory and temporary adhesive proteins in general appear to be large multidomain sequences, often with duplicated domains thought to facilitate protein–ligand interactions; (ii) both transitory and temporary bio-adhesives contain numerous glycosylated proteins; (iii) both transitory and temporary adhesives probably do not contain 3,4-dihydroxyphenyl-L-alanine (Dopa) (V. Kang 2020, unpublished data), which is more often associated with permanent adhesives from mussels, sandcastle worms and adult ascidians [89,90].

While none of the annotated limpet adhesive proteins share homology with known permanent adhesive proteins, *P-vulgata_5* aligned well (QC 91%, ID 27.70%) with a glyco-protein secreted by barnacle larvae for temporary adhesion, called the barnacle settlement-inducing protein complex (SIPC; NCBI Accession BAM28692.1). SIPC can adsorb to surfaces and is part of larval footprints [91,92]. This further supports the proposed adhesive function of *P-vulgata_5* and is another example of the similarity between limpet adhesives proteins and temporary adhesives. Indeed, proteins *P-vulgata_2, 4* and *6* share homology with known temporary adhesive proteins from flatworms (QEP99777.1 [51] and QAX24810.1 [54]) and sea stars (AHN92641.1 [24]). All these proteins also share similarities with the aforementioned group of proteins (fibrillin, zonadhesin, alpha-tectorin, IgGFc-binding protein). Moreover, a study examining the role of sulfated biopolymers in marine bio-adhesives also high-lighted the similarities between transitory and temporary adhesive secretions, where such moieties may serve a cohesive function in the adhesive material of limpets and sea stars but not in tubeworms or sea cucumbers [48]. Moreover, a review of adhesive secretions from marine invertebrates demonstrated that, based on comparisons between the amino acid composi-tions of whole adhesives, there are similarities between secretions from all species using non-permanent adhesion (i.e. temporary and transitory) [93].

Another animal that uses a similar type of transitory adhesion is the sea anemone (Actiniaria). Sea anemones com-monly occupy a similar ecological niche as limpets, and they can alternate between stationary and locomotive states [94]. One species, *Epiactis prolifera*, is capable of an average daily movement of 0.18 pedal disc diameter [95]. While the exact mechanism of pedal movement remains unclear, it seems likely that a combination of retrograde pedal waves and punctuated step-like movements is involved [95]. A recent transcriptomic study of the glass anemone *Exaiptasia pallida* has identified numerous upregulated genes in the pedal disc that may be important in bio-adhesion [56]. Enriched domains include protease inhibitors and metallopeptidases, analogous to the limpet adhesive proteins that may serve a role in defence or to transition the mucus from adhesive to locomotive, and protein–ligand binding domains, similar to both the limpet and temporary adhesives. Interestingly, the most abundantly expressed sequence in the pedal disc, Aip-gene2358, shared homology with *P-vulgata_10*, mainly from the tandem repeats of scavenger receptor cysteine-rich (SRCR) domains. Further studies are needed to ascertain the functional role of SRCR repeats in adhesive secretions, since SRCR participates in a wide range of activities, includ-ing ligand binding to lipoproteins and selected polyanions

[96,97], immune responses in marine invertebrates [73], and are also present in spider silk glands [98]. It should be noted, however, that although both limpets and sea ane-mones may participate in a similar type of adhesion and share related protein domains, there are no striking analogies that clearly differentiate transitory adhesive proteins from temporary adhesive ones.

## 5. Conclusion

The common limpet, *Patella vulgata*, has intrigued researchers for over a century with their impressive attachment strength. While previous studies have proposed both suction and glue-like attachment as mechanisms underlying limpet adhesion, we found only slight pressure differences gener-ated beneath the pedal sole of *P. vulgata* during both undisturbed locomotion and simulated predatory attacks. Based on the pressure recordings and behavioural obser-vations, we conclude that limpet pedal mucus is a bio-adhesive that provides a strong bond to the attachment surface. Our detailed analysis of the limpet pedal mucus has revealed novel insights into the molecular components of limpet bio-adhesive: (i) lectin staining assays confirmed the presence of several glycans specific to pedal sole glands and highlighted secretory granules; (ii) transcriptome-guided proteomics identified 171 adhesive protein candidates present in three types of limpet pedal mucus; (iii) *in situ* hybridization localized the expression of a selection of these proteins, four of which were present only at the pedal sole. Our annotation of pedal mucus protein sequences identified numerous domains often found in known temporary adhesives, along with multiple predicted sites for glycosyla-tion. Furthermore, these proteins are capable of protein–ligand interactions and are likely to oligomerize and cross-link to form a strong bio-adhesive. We also identified two protein architectures that have not been previously described in marine adhesive secretions: first is an SCO-spondin-like protein *P-vulgata_3*, which can potentially form fibres and raises interesting questions about the re-purposing of a highly conserved protein during evolution; second is a poten-tially potent defensive protein *P-vulgata_14*, with multiple domains for recognition and degradation of proteins and glycans. Although we have yet to identify key molecular differences between temporary and tidal transitory adhesives, our study is the first in-depth molecular character-ization of a model organism for tidal transitory adhesion and provides a solid foundation for future work.

Data accessibility. Encoding cDNA sequences of the 14 annotated pro-teins (*P-vulgata_1* to *14*) have been included in the electronic supplementary material. Raw sequencing reads and the assembled transcriptome has been deposited to the NCBI BioProject database under accession number PRJNA613775.

Authors' contributions. V.K., B.L. and P.F. conceived the study. V.K. col-lected the data for all experiments and drafted the manuscript. B.L. helped perform *in situ* hybridization, lectin assays and histochemical stainings. P.F. helped coordinate and provided guidance throughout the study. R.W. collected the proteomics data and helped interpret the results. All authors revised and approved the final manuscript.

Competing interests. The authors declare that they have no competing interests.

Funding. V.K. was funded by the European Union's Horizon 2020 research and innovation programme under the Marie Skłodowska-Curie grant agreement no. 642861. B.L. is funded by a Schrödinger Fellowship of the Austrian Science Fund (FWF): [J-4071]. P.F. is

Research Director of the Fund for Scientific Research of Belgium (FRS-FNRS). Work was partially supported by a Malacological Society of London Research Grant (awarded to V.K.), FNRS CDR Grant no J.0013.18, by the 'Communauté française de Belgique— Actions de Recherche Concertées' [ARC-17/21 UMONS 3], and by the European Cooperation in Science and Technology (COST) Action CA15216 (STSM no. 38594 and 41591). B.L., V.K. and P.F. are members of the COST Action 'European Network of Bioadhesion Expertise' (CA15216).

Acknowledgements. V.K. would like to thank Dr Walter Federle for his helpful feedback throughout the study. V.K. is grateful to Marie Bonneel and Morgane Algrain for their help with some of the molecular biology experiments, and to Jérôme Delroisse for his help with transcriptomic analyses. We thank Thomas Ostermann and Peter Ladurner (University of Innsbruck) for access to their BLAST server and the UMONS Bioprofiling platform, Cyril-Terence Mascolo and Corentin Decroo for their advice on mass spectrometry experiments.

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
