## [Reviewer comments · Open Biology]

Review History

RSOB-20-0019.R0 (Original submission)

Review form: Reviewer 1

Recommendation

Major revision is needed (please make suggestions in comments)

Do you have any ethical concerns with this paper?

No

Comments to the Author

The research article is well-written. The research explores adhesive-mucus production in a common limpet species. I enjoyed reading the article and it appears the limpet is molecularly similar to many bio-adhesive marine invertebrates whilst exhibiting several notable differences. I do have a few concerns (some major) which I encourage the authors to address with satisfactory answers when resubmitting for further review. With re-writing and clarification (a major revision), I believe this manuscript has merit to be published as a suitable resource for future research into gastropod and marine adhesion biosystems. Major and minor critics are listed below.

Major revisions and points of clarification:

Line 180-181: To my understanding, limpets were left for 30 minutes outside circulating tank

water in a humid environment to produce 'secondary mucus'. Is mucus production / composition affected by entailing environment (non-aqueous vs aqueous)? Are difference in primary mucus and secondary mucus a result of difference in environment? Would it have been more appropriate to subject the limpets to submersion before secondary mucus was collected. Although Dr. Smith has proposed various types of mucus in the past, I ask the authors why they didn't conduct rheology investigation into the mucus types within *P. vulgata*. I would have liked to have seen this to show that in fact the mucus types are compositionally different. Are differences in total protein and carbohydrate content different, as seen in the species, *L. limatula*? This would be a good addition.

Line 222-232: I am surprised that the authors have used several pedal sole dissections from only one individual animal (n=1) to form a de novo transcriptome. The most biologically relevant and sound approach in order to capture transcript variants and a more complete de novo transcriptome would be to use several individual animal soles as conducted in most other studies. Perhaps a more complete transcriptome would allow the authors to identify more adhesion candidates. I therefore request the authors to state this potential pitfall in the discussion and clarify why they only used one animal. As a note, without the proteomic validation I would have rejected the paper.

Additionally, no RIN numbers for RNA samples have been provided, RIN numbers for sequencing need to be approx. 7 for non-degraded RNA. A few lines describe the bioinformatics conducted. I am not satisfied with this. What quality checks and trimming was completed on the reads (with settings) pre-assembly? Also state the version of Trinity.

Additionally on line 389, authors discuss FPKM expression values - I would therefore request information on read alignment software and settings used, as well as software used to conduct read counts. Furthermore on line 366, the authors state 'a high-quality transcriptome' - given that the assembly was of de novo origin (without genomic reference), I ask what method was used to clarify it was of 'high-quality'. I therefore ask for further analysis to be conducted such as BUSCO or similar orthology assignment completeness. If not provided, remove the descriptive 'high quality'. All this information and further quality checks can be obtained by contacting Beijing Genomic Institute, China.

I would like to know what criteria the authors used for down-selection of candidates for ISH screening. The ISH protocol description on lines 260-263 is disappointing given the technicalities of the method. I request further details on preparation, permeabilization, wash, hybridization and stringency wash steps. The authors need to think about reproducibility - if the reader wished to complete the work again. Additionally, I see no images of negative controls. What did you use as controls? Additionally, the paragraph on background staining in the supplementary needs to be expanded in more detail. Did the lead author visit a collaborating lab or before they left the lab in Cambridge? Somewhat confusing.

Supplementary Figure 2: You have a ladder and a size estimation breakdown on the opposite side, this is not stated in the legend for the reader to understand what you want to portray. I presume the original ladder was out of scale? Supplementary Figure 2b legend does not read correctly, please amend. No mention of asialofetuin's role in the image - Obviously a glycoprotein from calf serum as a reference, but not stated in 2c legend. Perhaps unclear to some readers?

Minor comments:

Line 44: remove the word 'but' - isn't required.

Line 47: Grazers, missing the 's' on Grazer. Biofilm yes, but they also graze on algae and detritus. Figure 1: Nice figure showing how strong limpet adhesion actually is. It's clear to me you are holding the rock suspended via the limpet. Can you modify the legend to describe this more clearly? It may not be clear to others.

Line 95 re-write as 'remains limited compared to our understanding of other marine bio-adhesive secretions'

Lines 96-98: remove 'decode' and re-write. Perhaps 'assemble and analyse transcriptomes and proteomes to characterise the molecular networks which govern bio-adhesive systems' is more appropriate here?

Line 111: 'Modern molecular biology tools' - change to 'a range of appropriate molecular biology approaches'. Some techniques are now 10-20 years plus in use.

Line 115: What was 'careful' about it?

Line 165: Replace two consecutive bracketed text instances with one, using a semi-colon.

Line 163-181: The circulating sea water and the ASW were of the same salinity? This isn't stated.

Line 199: β -MSH = β -melanocyte-stimulating hormone in existing literature. Perhaps abbreviations, 2BME or 2 β ME are more appropriate?

Line 318 – Authors may have over-sighted. Please add 'approximately 60%'. On measuring, it appears that pressures compared are -0.60 kPA and -0.97kPA. Not exactly -0.60 kPA and -1.00 kPA.

Line 319: Add perpendicularly from the surface, as written in line 310?

Line 327: I'd recommend 'protein bands' rather than 'a few proteins', cannot distinguish separate proteins from the smears, nor quantify. I see it is correctly done in line 331.

Line 332- Authors refer to different mucus as IPAM, BPAM, SAM. Supplementary figure 2 refers to 'old mucus, fresh mucus, footprint'. Confusing. Fix. I presume footprint is referring to IPAM?

Line 347: LCA – first time this abbreviation has been stated – please write in full. Same for all lectin stains.

Lines 370-371: Did you manually identify these with knowledge and literature searches with known adhesives? Unclear how you identified. Line 374 – what 'stringent' criteria?

Table 1: Don't assume readers know each lectin stain name. Full name can be provided in the legend or in the table.

Line 382: What database did you use for manual annotation of conserved protein domains and what search criteria was used? There are several available.

Line 465 – state what species lysozyme C is from. I've checked and it's *canis lupus familiaris*. State it. Same with line 469 – *Homo sapiens*. Check throughout for others. Be consistent.

Line 625 –The following statement is quite bold– “*P. vulgata*_14 is the first example of an annotated protein from marine bio-adhesive with multiple domains relevant for antibacterial activity”. Several papers on echinoderms, anemones and others highlight proteins potentially associated with antimicrobial activity / immunity / defence. Perhaps change the wording as wet-lab verification work needs completed to assign a function to the protein. It well could have a different role?

Line 685-686: No tyrosinase like orthologs within the transcriptome? If not, you could add this to this section.

Supplementary Table 2 – there is a bracket in the legend, remove.

Supplementary table 3: What does NA mean? – I presume these are the ones that failed in ISH?

Supplementary figure 3: legend and alignments on one page please.

Supplementary: “DNA sequences of the fourteen annotated protein” – please change to “Encoding cDNA” as RNA-seq captures RNA which is then reverse transcribed to cDNA for sequencing, not gDNA which also includes introns – I assume a simple over-assumption by authors. Line 762 as well.

Additional data: Please publish the assembled transcriptome with raw reads to NCBI. It will be a good resource for the bio-adhesive community.

I look forward to receiving the resubmitted version.

Review form: Reviewer 2

Recommendation

Accept with minor revision (please list in comments)

Do you have any ethical concerns with this paper?

No

Comments to the Author

This is an interesting and valuable contribution. The work is well-executed and thoroughly analyzed. It will be a significant contribution to the bioadhesion literature. I only have some minor comments.

There are large differences in the abundance of the different transcripts based on the FPKM data. This seems very important and should be discussed. The relative abundance gives some insight into the role of the proteins. I would expect something that is extremely abundant to be more likely to be a major structural component of the secretion, whereas something that is much less abundant may serve a more specialized role, or a catalytic role.

Related to the previous point, what were the criteria for selecting the fourteen sequences for more detailed analysis? Some seem to be very low abundance, based on the FPKM data, and some lack signal peptides suggesting that they aren't secreted. Without knowing the selection criteria, it is hard to be confident that all the main proteins in the glue are represented in the fourteen selected. It does seem that the authors succeeded in capturing the main proteins, since some of the most abundant transcripts from among the fourteen do have sizes that might line up with the proteins seen in SDS-PAGE. Nevertheless, more information on the selection criteria would help.

How was the pressure sensor calibrated? The authors should provide evidence that the measurements are accurate.

Line 54 should be qualified. Instead of "the mechanisms responsible for the limpets' strong attachment remain unresolved", it should read "the mechanisms responsible for patellid limpets' strong attachment remain unresolved". As the authors make clear in their review of the literature, the mechanism for lottiid limpets seems relatively well-resolved.

In Fig 3c, it would be helpful to know when the force was applied, and when the limpet detached. Did the sudden drop in pressure correspond with a sudden increase in detaching force, or was the force applied for some time and the pressure only dropped at the moment of failure? The latter would be further evidence of gluing. It might be worth noting that a glued animal might produce the sudden pressure drop after adhesion fails and the foot begins to deform and pull away. In Smith's paper on lottiid limpets, it was noted that this effect can create a transient pressure drop of about 8 kPa. This is comparable to what the authors see in Fig 3c.

In Supplementary Fig.2, the authors refer to footprint mucus and adhesive mucus. They should use their terminology of IPAM, BPAM and SAM throughout the paper.

On line 260, the authors state that samples were fixed in paraformaldehyde. Were they then embedded in paraffin?

Review form: Reviewer 3**Recommendation**

Accept as is

Do you have any ethical concerns with this paper?

No

Comments to the Author

The paper is a good start to understanding limpet adhesion. It provides a catalog of molecular parts that will be necessary for more mechanistic studies.

The work is introduced well and presented clearly.

Decision letter (RSOB-20-0019.R0)

09-Mar-2020

Dear Mr Kang,

We are writing to inform you that the Editor has reached a decision on your manuscript RSOB-20-0019 entitled "Molecular insights into the powerful mucus-based adhesion of limpets (*Patella vulgata* L.)", submitted to Open Biology.

As you will see from the reviewers' comments below, there are a number of criticisms that prevent us from accepting your manuscript at this stage. The reviewers suggest, however, that a revised version could be acceptable, if you are able to address their concerns. If you think that you can deal satisfactorily with the reviewer's suggestions, we would be pleased to consider a revised manuscript.

The revision will be re-reviewed, where possible, by the original referees. As such, please submit the revised version of your manuscript within four weeks. If you do not think you will be able to meet this date please let us know immediately.

When submitting your revised manuscript, please respond to the comments made by the referee(s) and upload a file "Response to Referees" in "Section 6 - File Upload". You can use this to document any changes you make to the original manuscript. In order to expedite the processing of the revised manuscript, please be as specific as possible in your response to the referee(s).

Please see our detailed instructions for revision requirements
<https://royalsociety.org/journals/authors/author-guidelines/>

Sincerely,
The Open Biology Team
<mailto:openbiology@royalsociety.org>

Reviewer(s)' Comments to Author(s):

Referee: 1

Comments to the Author(s)

The research article is well-written. The research explores adhesive-mucus production in a common limpet species. I enjoyed reading the article and it appears the limpet is molecularly

similar to many bio-adhesive marine invertebrates whilst exhibiting several notable differences. I do have a few concerns (some major) which I encourage the authors to address with satisfactory answers when resubmitting for further review. With re-writing and clarification (a major revision), I believe this manuscript has merit to be published as a suitable resource for future research into gastropod and marine adhesion biosystems. Major and minor critics are listed below.

Major revisions and points of clarification:

Line 180-181: To my understanding, limpets were left for 30 minutes outside circulating tank water in a humid environment to produce 'secondary mucus'. Is mucus production / composition affected by entailing environment (non-aqueous vs aqueous)? Are difference in primary mucus and secondary mucus a result of difference in environment? Would it have been more appropriate to subject the limpets to submersion before secondary mucus was collected. Although Dr. Smith has proposed various types of mucus in the past, I ask the authors why they didn't conduct rheology investigation into the mucus types within *P. vulgata*. I would have liked to have seen this to show that in fact the mucus types are compositionally different. Are differences in total protein and carbohydrate content different, as seen in the species, *L. limatula*? This would be a good addition.

Line 222-232: I am surprised that the authors have used several pedal sole dissections from only one individual animal (n=1) to form a de novo transcriptome. The most biologically relevant and sound approach in order to capture transcript variants and a more complete de novo transcriptome would be to use several individual animal soles as conducted in most other studies. Perhaps a more complete transcriptome would allow the authors to identify more adhesion candidates. I therefore request the authors to state this potential pitfall in the discussion and clarify why they only used one animal. As a note, without the proteomic validation I would have rejected the paper.

Additionally, no RIN numbers for RNA samples have been provided, RIN numbers for sequencing need to be approx. 7 for non-degraded RNA. A few lines describe the bioinformatics conducted. I am not satisfied with this. What quality checks and trimming was completed on the reads (with settings) pre-assembly? Also state the version of Trinity.

Additionally on line 389, authors discuss FPKM expression values – I would therefore request information on read alignment software and settings used, as well as software used to conduct read counts. Furthermore on line 366, the authors state 'a high-quality transcriptome' – given that the assembly was of de novo origin (without genomic reference), I ask what method was used to clarify it was of 'high-quality'. I therefore ask for further analysis to be conducted such as BUSCO or similar orthology assignment completeness. If not provided, remove the descriptive 'high quality'. All this information and further quality checks can be obtained by contacting Beijing Genomic Institute, China.

I would like to know what criteria the authors used for down-selection of candidates for ISH screening. The ISH protocol description on lines 260-263 is disappointing given the technicalities of the method. I request further details on preparation, permeabilization, wash, hybridization and stringency wash steps. The authors need to think about reproducibility - if the reader wished to complete the work again. Additionally, I see no images of negative controls. What did you use as controls? Additionally, the paragraph on background staining in the supplementary needs to be expanded in more detail. Did the lead author visit a collaborating lab or before they left the lab in Cambridge? Somewhat confusing.

Supplementary Figure 2: You have a ladder and a size estimation breakdown on the opposite side, this is not stated in the legend for the reader to understand what you want to portray. I presume the original ladder was out of scale? Supplementary Figure 2b legend does not read correctly, please amend. No mention of asialofetuin's role in the image - Obviously a glycoprotein from calf serum as a reference, but not stated in 2c legend. Perhaps unclear to some readers?

Minor comments:

Line 44: remove the word 'but' – isn't required.

Line 47: Grazers, missing the 's' on Grazer. Biofilm yes, but they also graze on algae and detritus.

Figure 1: Nice figure showing how strong limpet adhesion actually is. It's clear to me you are holding the rock suspended via the limpet. Can you modify the legend to describe this more clearly? It may not be clear to others.

Line 95 re-write as 'remains limited compared to our understanding of other marine bio-adhesive secretions'

Lines 96-98: remove 'decode' and re-write. Perhaps 'assemble and analyse transcriptomes and proteomes to characterise the molecular networks which govern bio-adhesive systems' is more appropriate here?

Line 111: 'Modern molecular biology tools' - change to 'a range of appropriate molecular biology approaches'. Some techniques are now 10-20 years plus in use.

Line 115: What was 'careful' about it?

Line 165: Replace two consecutive bracketed text instances with one, using a semi-colon.

Line 163-181: The circulating sea water and the ASW were of the same salinity? This isn't stated.

Line 199: β -MSH = β -melanocyte-stimulating hormone in existing literature. Perhaps abbreviations, 2BME or 2 β ME are more appropriate?

Line 318 - Authors may have over-sighted. Please add 'approximately 60%'. On measuring, it appears that pressures compared are -0.60 kPA and -0.97kPA. Not exactly -0.60 kPA and -1.00 kPA.

Line 319: Add perpendicularly from the surface, as written in line 310?

Line 327: I'd recommend 'protein bands' rather than 'a few proteins', cannot distinguish separate proteins from the smears, nor quantify. I see it is correctly done in line 331.

Line 332- Authors refer to different mucus as IPAM, BPAM, SAM. Supplementary figure 2 refers to 'old mucus, fresh mucus, footprint'. Confusing. Fix. I presume footprint is referring to IPAM?

Line 347: LCA - first time this abbreviation has been stated - please write in full. Same for all lectin stains.

Lines 370-371: Did you manually identify these with knowledge and literature searches with known adhesives? Unclear how you identified. Line 374 - what 'stringent' criteria?

Table 1: Don't assume readers know each lectin stain name. Full name can be provided in the legend or in the table.

Line 382: What database did you use for manual annotation of conserved protein domains and what search criteria was used? There are several available.

Line 465 - state what species lysozyme C is from. I've checked and it's *canis lupus familiaris*. State it. Same with line 469 - *Homo sapiens*. Check throughout for others. Be consistent.

Line 625 -The following statement is quite bold- "P. vulgata_14 is the first example of an annotated protein from marine bio-adhesive with multiple domains relevant for antibacterial activity". Several papers on echinoderms, anemones and others highlight proteins potentially associated with antimicrobial activity / immunity / defence. Perhaps change the wording as wet-lab verification work needs completed to assign a function to the protein. It well could have a different role?

Line 685-686: No tyrosinase like orthologs within the transcriptome? If not, you could add this to this section.

Supplementary Table 2 - there is a bracket in the legend, remove.

Supplementary table 3: What does NA mean? - I presume these are the ones that failed in ISH?

Supplementary figure 3: legend and alignments on one page please.

Supplementary: "DNA sequences of the fourteen annotated protein" - please change to "Encoding cDNA" as RNA-seq captures RNA which is then reverse transcribed to cDNA for sequencing, not gDNA which also includes introns - I assume a simple over-assumption by authors. Line 762 as well.

Additional data: Please publish the assembled transcriptome with raw reads to NCBI. It will be a good resource for the bio-adhesive community.

I look forward to receiving the resubmitted version.

Referee: 2

Comments to the Author(s)

This is an interesting and valuable contribution. The work is well-executed and thoroughly analyzed. It will be a significant contribution to the bioadhesion literature. I only have some minor comments.

There are large differences in the abundance of the different transcripts based on the FPKM data. This seems very important and should be discussed. The relative abundance gives some insight into the role of the proteins. I would expect something that is extremely abundant to be more likely to be a major structural component of the secretion, whereas something that is much less abundant may serve a more specialized role, or a catalytic role.

Related to the previous point, what were the criteria for selecting the fourteen sequences for more detailed analysis? Some seem to be very low abundance, based on the FPKM data, and some lack signal peptides suggesting that they aren't secreted. Without knowing the selection criteria, it is hard to be confident that all the main proteins in the glue are represented in the fourteen selected. It does seem that the authors succeeded in capturing the main proteins, since some of the most abundant transcripts from among the fourteen do have sizes that might line up with the proteins seen in SDS-PAGE. Nevertheless, more information on the selection criteria would help.

How was the pressure sensor calibrated? The authors should provide evidence that the measurements are accurate.

Line 54 should be qualified. Instead of "the mechanisms responsible for the limpets' strong attachment remain unresolved", it should read "the mechanisms responsible for patellid limpets' strong attachment remain unresolved". As the authors make clear in their review of the literature, the mechanism for lottiid limpets seems relatively well-resolved.

In Fig 3c, it would be helpful to know when the force was applied, and when the limpet detached. Did the sudden drop in pressure correspond with a sudden increase in detaching force, or was the force applied for some time and the pressure only dropped at the moment of failure? The latter would be further evidence of gluing. It might be worth noting that a glued animal might produce the sudden pressure drop after adhesion fails and the foot begins to deform and pull away. In Smith's paper on lottiid limpets, it was noted that this effect can create a transient pressure drop of about 8 kPa. This is comparable to what the authors see in Fig 3c.

In Supplementary Fig.2, the authors refer to footprint mucus and adhesive mucus. They should use their terminology of IPAM, BPAM and SAM throughout the paper.

On line 260, the authors state that samples were fixed in paraformaldehyde. Were they then embedded in paraffin?

Referee: 3

Comments to the Author(s)

The paper is a good start to understanding limpet adhesion. It provides a catalog of molecular parts that will be necessary for more mechanistic studies.

The work is introduced well and presented clearly.

Author's Response to Decision Letter for (RSOB-20-0019.R0)

See Appendix A.

RSOB-20-0019.R1 (Revision)

Review form: Reviewer 1

Recommendation

Accept with minor revision (please list in comments)

Do you have any ethical concerns with this paper?

No

Comments to the Author

All changes that have been made are now to my satisfaction. The article is now in better shape. Only two minor revisions to make. Well done and all the best for the future.

Review form: Reviewer 2

Recommendation

Accept as is

Do you have any ethical concerns with this paper?

No

Comments to the Author

The authors have addressed the concerns adequately.

I have one minor suggestion the authors might choose to address if they want. For selecting the subset of proteins to analyze, they said that they "limited our selection to proteins that were ranked highly by ProteinPilot". It would be useful to indicate what constitutes being ranked highly. Is there a threshold that they could provide, or anything more quantitative?

Decision letter (RSOB-20-0019.R1)

27-Apr-2020

Dear Mr Kang

We are pleased to inform you that your manuscript RSOB-20-0019.R1 entitled "Molecular insights into the powerful mucus-based adhesion of limpets (*Patella vulgata* L.)" has been accepted by the Editor for publication in *Open Biology*. The reviewer(s) have recommended publication, but also suggest some minor revisions to your manuscript. Therefore, we invite you to respond to the reviewer(s)' comments and revise your manuscript.

Please submit the revised version of your manuscript within 7 days. If you do not think you will be able to meet this date please let us know immediately and we can extend this deadline for you.

- 1) A text file of the manuscript (doc, txt, rtf or tex), including the references, tables (including captions) and figure captions. Please remove any tracked changes from the text before submission. PDF files are not an accepted format for the "Main Document".
- 2) A separate electronic file of each figure (tiff, EPS or print-quality PDF preferred). The format should be produced directly from original creation package, or original software format. Please note that PowerPoint files are not accepted.
- 3) Electronic supplementary material: this should be contained in a separate file from the main text and meet our ESM criteria (see <http://royalsocietypublishing.org/instructions-authors#question5>). All supplementary materials accompanying an accepted article will be treated as in their final form. They will be published alongside the paper on the journal website and posted on the online figshare repository. Files on figshare will be made available approximately one week before the accompanying article so that the supplementary material can be attributed a unique DOI.

Online supplementary material will also carry the title and description provided during submission, so please ensure these are accurate and informative. Note that the Royal Society will not edit or typeset supplementary material and it will be hosted as provided. Please ensure that the supplementary material includes the paper details (authors, title, journal name, article DOI). Your article DOI will be 10.1098/rsob.2016[last 4 digits of e.g. 10.1098/rsob.20160049].

- 4) A media summary: a short non-technical summary (up to 100 words) of the key findings/importance of your manuscript. Please try to write in simple English, avoid jargon, explain the importance of the topic, outline the main implications and describe why this topic is newsworthy.

Images

Data-Sharing

It is a condition of publication that data supporting your paper are made available. Data should be made available either in the electronic supplementary material or through an appropriate repository. Details of how to access data should be included in your paper. Please see <https://royalsociety.org/journals/authors/author-guidelines/> for more details.

Data accessibility section

Sincerely,
The Open Biology Team
<mailto:openbiology@royalsociety.org>

Reviewer(s)' Comments to Author:

Referee: 2

Comments to the Author(s)

The authors have addressed the concerns adequately.

I have one minor suggestion the authors might choose to address if they want. For selecting the subset of proteins to analyze, they said that they "limited our selection to proteins that were ranked highly by ProteinPilot". It would be useful to indicate what constitutes being ranked highly. Is there a threshold that they could provide, or anything more quantitative?

Referee: 1

Comments to the Author(s)

All changes that have been made are now to my satisfaction. The article is now in better shape. Only two minor revisions to make. Well done and all the best for the future.

Decision letter (RSOB-20-0019.R2)

14-May-2020

Dear Mr Kang

We are pleased to inform you that your manuscript entitled "Molecular insights into the powerful mucus-based adhesion of limpets (*Patella vulgata* L.)" has been accepted by the Editor for publication in Open Biology.

Article processing charge

Please note that the article processing charge is immediately payable. A separate email will be sent out shortly to confirm the charge due. The preferred payment method is by credit card; however, other payment options are available.

Sincerely,

The Open Biology Team
mailto: openbiology@royalsociety.org

Appendix A

Referee 1

We would like to express our gratitude to Referee 1 for providing thorough and insightful suggestions that have undoubtedly improved our manuscript. Please see our response to individual points below.

Major revisions and points of clarification:

1. Line 180-181: To my understanding, limpets were left for 30 minutes outside circulating tank water in a humid environment to produce 'secondary mucus'. Is mucus production / composition affected by entailing environment (non-aqueous vs aqueous)? Are difference in primary mucus and secondary mucus a result of difference in environment? Would it have been more appropriate to subject the limpets to submersion before secondary mucus was collected.
 - 1.a. We thank the reviewer for pointing out that our collection method may influence the nature of the secondary adhesive mucus. We acknowledge this possibility and have added the following to the Methods section (now lines 185-187): *It should be noted that as SAM was collected in air, the composition may vary if sampled from individuals left in an aqueous environment.*
 - 1.b. Please note that we developed our collection technique to minimise contamination from different individuals and mucus from other parts of the body, which we believe is more difficult to control in an aqueous environment than in air.
2. Although Dr. Smith has proposed various types of mucus in the past, I ask the authors why they didn't conduct rheology investigation into the mucus types within *P. vulgata*. I would have liked to have seen this to show that in fact the mucus types are compositionally different. Are differences in total protein and carbohydrate content different, as seen in the species, *L. limatula*? This would be a good addition.
 - 2.a. We agree that rheology can be a useful tool to extract bulk material properties from biological fluids if sufficient amount of the material is available. Although Grenon & Walker have previously attempted to characterise the rheology of *P. vulgata* pedal sole mucus (similar to our secondary adhesive mucus), they had to pool samples from numerous individuals to have sufficient volume for bulk rheology (Grenon & Walker 1980 Comp Biochem Physiol. 66B:451–458). We strongly believe that secreted mucus from different individuals cannot be combined to form a representative sample because we observed pooled mucus to be inhomogeneous, and we do not expect internal molecular networks to readily combine post-secretion and across samples. Hence, as pooling mucus will not accurately represent mucus material properties, and since individual limpets produce extremely limited quantities of each mucus type, we do not think that rheology would be a feasible technique to distinguish mucus types. That being said, we did observe qualitative differences between the mucus types, which may be of interest to the reader and have added to the Results section (lines 347-352): *We observed a number of qualitative differences between the types of mucus: first, IPAM was a thin layer left on the surface when the limpet was detached that sometimes felt like a raised solid patch. The thin layer of IPAM became visible with crystal violet staining. BPAM, on the other hand, was visible as an opaque swollen layer on top of the pedal sole and could at times be removed as an intact sheet of mucus. Lastly, the small quantities of SAM produced on the pedal sole easily broke apart during collection and did not form sheets like BPAM.*
 - 2.b. We acknowledge that analysis of total protein and carbohydrate content would have provided additional insight into the different types of mucus. Unfortunately, we did not consider this when we had study animals on hand. Since limpet mucus carbohydrate and protein content vary between individual size and season (Davies et al 1990 JEMBE 144: 101-112), we are unable to obtain this information retroactively. However, we hope that the additional qualitative description of the three mucus types outlined above would be useful to the reviewer and the wider readership.

3. Line 222-232: I am surprised that the authors have used several pedal sole dissections from only one individual animal (n=1) to form a de novo transcriptome. The most biologically relevant and sound approach in order to capture transcript variants and a more complete de novo transcriptome would be to use several individual animal soles as conducted in most other studies. Perhaps a more complete transcriptome would allow the authors to identify more adhesion candidates. I therefore request the authors to state this potential pitfall in the discussion and clarify why they only used one animal. As a note, without the proteomic validation I would have rejected the paper.
 - 3.a. When we were devising the project, we decided to focus our analysis on the adhesive mucus proteome as we were confident that we could collect sufficient material for proteomics. The pedal sole transcriptome was generated to map the proteomic data to help us identify more complete candidate sequences. Adhesive proteins are often large and repetitive (Hennebert et al., 2014, Wunderer et al., 2019), and are therefore difficult to assemble using short-read transcriptomics (Lengerer et al., 2018, Pjeta et al., 2020). Since pooling multiple individuals for the transcriptome would likely introduce additional variation at the individual level and consequently increase the variation of each transcript, we used a single individual in an effort to obtain longer assembled transcripts. This decision was influenced by our prior experience identifying adhesive protein candidates from flatworms and sea stars. In flatworms, transcriptomes based on pooled individuals led to incomplete assembly of adhesive transcripts, which caused an overestimation of the number of adhesive proteins (e.g., ten separate transcripts turned out to be fragments of two large adhesive transcripts; see Lengerer et al., 2018, Wunderer et al. 2019). In sea stars, the long full-length transcript of sea star footprint protein 1 (Sfp-1) was found as one assembled transcript in the *de novo* transcriptome from a single individual and mapped using proteomic data (Hennebert et al, 2014, 2015). Hence, we opted to use a single limpet for the transcriptome. However, we agree that an explanation for our decision to use a single individual and the limitations thereof will improve the manuscript. As such, we have added the following text to the Discussion (lines 613-619): *It is worth mentioning that the transcriptome was based on the sequencing data from the pedal sole of a single limpet specimen. Since adhesive proteins are often large and repetitive [24, 54], they tend to be inadequately assembled with short-read transcriptomics [61, 62]. We sought to increase mapped transcript lengths by reducing the complexity of the input RNA and minimising transcript variation caused by pooling samples from multiple individuals. Although our analysis showed that the transcriptome is of good quality, we want to highlight that due to the limited sample size, some transcripts and transcript variations may not be represented in this dataset.*
 - 3.b. Moreover, we have amended all references to the 'high quality' of the transcriptome (see 5a below).
4. Additionally, no RIN numbers for RNA samples have been provided, RIN numbers for sequencing need to be approx. 7 for non-degraded RNA. A few lines describe the bioinformatics conducted. I am not satisfied with this. What quality checks and trimming was completed on the reads (with settings) pre-assembly? Also state the version of Trinity.
 - 4.a. We have expanded the Methods and added further details to include the quality checks (including RIN) and pre-assembly trimming conducted by Beijing Genomics Institute. Lines 234-240 in the main text now reads: *Subsequent sequencing, data processing, and transcriptome assembly were performed at the Beijing Genomic Institute, China (BGI). Integrity of the isolated RNA was assessed by gel electrophoresis and via Agilent Bioanalyser prior to sequencing (RNA integrity number: 6). Illumina HiSeqXTen platform was used to generate 150 bp paired-end reads, and the raw reads were filtered to remove adaptors and low-quality reads (defined internally within BGI as reads whereupon the percentage of bases with a quality score less than 10 was greater than 20%). Cleaned reads were used for de novo assembly of the transcriptome with Trinity software v2.0.6 [28] and assembled into Unigenes with Tgicl v2.0.6 [29].*
5. Additionally on line 389, authors discuss FPKM expression values – I would therefore request information on read alignment software and settings used, as well as software used to conduct

read counts. Furthermore on line 366, the authors state ‘a high-quality transcriptome’ – given that the assembly was of de novo origin (without genomic reference), I ask what method was used to clarify it was of ‘high-quality’. I therefore ask for further analysis to be conducted such as BUSCO or similar orthology assignment completeness. If not provided, remove the descriptive ‘high quality’. All this information and further quality checks can be obtained by contacting Beijing Genomic Institute, China.

- 5.a. We have revised the relevant Methods section to include more details on the alignment and read count estimates (now lines 236-243): *Illumina HiSeqXTen platform was used to generate 150 bp paired-end reads, and the raw reads were filtered to remove adaptors and low-quality reads (see Supplementary Materials for more details). Cleaned reads were used for de novo assembly of the transcriptome with Trinity software v2.0.6 [28] and assembled into Unigenes with Tgicl v2.0.6 [29]. Fragments per kilobase of transcript per million mapped reads (FPKM) values were calculated by first mapping clean reads to Unigenes with bowtie2 [30] (v2.2.5, sensitive mode; see Supplementary Materials for full software settings), then RSEM [31] (v1.2.12, default parameters) was used to quantify expression levels.*
- 5.b. Additionally, we have added more details on the pre-assembly trimming and the full software parameters for alignment using *bowtie2* in Supplementary Materials (pg 1).
- 5.c. BUSCO analyses were performed to evaluate the completeness of the assembled Unigenes. We have revised the relevant Methods section to include details about the analysis (lines 243-250): *To assess transcriptome assembly and annotation completeness, we conducted an analysis based on the Benchmarking Universal Single-Copy Orthologs (BUSCO) using BUSCO v3.0.2 [32] for metazoa_odb9 and eukaryote_odb9 datasets. Based on the metazoan dataset, the assembled transcriptome was estimated to be 91.5% complete with 894 complete BUSCOs, 4.4% (43) fragmented BUSCOs and 4.1% (41) missing BUSCOs from a total of 978 BUSCO groups searched. Similar values were obtained with the eukaryota dataset. Note that these BUSCO numbers are in line with those from the *Lottia gigantea* reference genome [32]. Raw sequencing reads and the assembled transcriptome has been deposited to the NCBI BioProject database under accession number PRJNA613775.*
6. I would like to know what criteria the authors used for down-selection of candidates for ISH screening. The ISH protocol description on lines 260-263 is disappointing given the technicalities of the method. I request further details on preparation, permeabilization, wash, hybridization and stringency wash steps. The authors need to think about reproducibility - if the reader wished to complete the work again. Additionally, I see no images of negative controls. What did you use as controls? Additionally, the paragraph on background staining in the supplementary needs to be expanded in more detail. Did the lead author visit a collaborating lab or before they left the lab in Cambridge? Somewhat confusing.
- 6.a. We acknowledge that more information on the ISH selection criteria is necessary, and have revised the main text to read (line 271-277): *From the combined list of candidate proteins, a subset was selected for further analysis using in situ hybridisation (ISH) based on the following criteria: first, we limited our selection to proteins that were ranked highly by ProteinPilot to ensure we were targeting proteins that were present in adhesive mucus. Second, we included candidates with conserved protein domains that were commonly associated with marine bio-adhesives (e.g., vWFD, EGF, lectins). Finally, we sought to sample proteins across the different types of mucus with the goal of identifying candidates associated with specific types of mucus (IPAM, BPAM, and SAM).*
- 6.b. We added a detailed description of the *in situ* hybridisation protocol to the Supplementary Materials. We also rephrased and expanded the explanation of the background staining (pg 8-9 and Supplementary Figure 5). To clarify, all the molecular work was conducted in Flammang’s group at the University of Mons, Belgium, during several extended collaboration trips undertaken by the lead author.
7. Supplementary Figure 2: You have a ladder and a size estimation breakdown on the opposite side, this is not stated in the legend for the reader to understand what you want to portray. I

presume the original ladder was out of scale? Supplementary Figure 2b legend does not read correctly, please amend. No mention of asialofetuin's role in the image - Obviously a glycoprotein from calf serum as a reference, but not stated in 2c legend. Perhaps unclear to some readers?

- 7.a. We are grateful for these suggestions and have implemented the following changes in Supplementary Figure 2 (now Supplementary Figure 3; pg 2): *SDS-PAGE gels of protein extracts from P. vulgata pedal sole mucus stained for additional information. BPAM: bulk primary adhesive mucus; SAM: secondary adhesive mucus; IPAM: interfacial primary adhesive mucus. (a) Coomassie Blue stain identified at least 11 prominent protein bands, ranging from ~40 to greater than 250 kDa. Note the approximate protein band sizes to the right of the gel image. (b) Smearing purple bands from PAS staining confirms that the presence of glycosylation for specific proteins within IPAM and not in BPAM. The strong smearing at the top of the gel indicates large complexes that failed to properly migrate into the gel. Asialo-fetuin from bovine serum used to as a reference glycoprotein to illustrate smearing pattern. (c) Multi-coloured bands from Stains-All highlight several differences between BPAM and IPAM (blue for highly acidic proteins and Ca²⁺-binding proteins, purple for intact proteoglycans, and pink for weakly acidic proteins).*

Minor comments:

1. Line 44: remove the word 'but' – isn't required.
 - 1.a. Addressed.
2. Line 47: Grazers, missing the 's' on Grazer. Biofilm yes, but they also graze on algae and detritus.
 - 2.a. We have revised the sentence as follows (line 45-48): *However, unlike adult mussels and barnacles that rely on filter-feeding and permanently adhere to surfaces in the intertidal zone, limpets are active grazers of biofilm and detritus [10]; hence, they can travel considerable distances while feeding (up to 1.5 m [6]).*
3. Figure 1: Nice figure showing how strong limpet adhesion actually is. It's clear to me you are holding the rock suspended via the limpet. Can you modify the legend to describe this more clearly? It may not be clear to others.
 - 3.a. We agree that the caption for Figure 1 could be clearer and have revised it to read: *Limpets (Patella vulgata) have evolved powerful attachments to withstand crashing tidal waves and predatory attacks. Here, one of the authors lifted a heavy rock by hooking onto a single limpet.*
4. Line 95 re-write as 'remains limited compared to our understanding of other marine bio-adhesive secretions'
 - 4.a. Addressed, lines 95-97: *While these earlier efforts offer initial biochemical descriptions of the limpet pedal mucus, our knowledge of its molecular components and their function remains limited compared to our understanding of other marine bio-adhesive secretions.*
5. Lines 96-98: remove 'decode' and re-write. Perhaps 'assemble and analyse transcriptomes and proteomes to characterise the molecular networks which govern bio-adhesive systems' is more appropriate here?
 - 5.a. We thank the reviewer for this suggestion and have modified the sentence as follows (lines 97-99): *Advances in sequencing technology and bioinformatics have allowed researchers to assemble and analyse transcriptomes and proteomes in order to characterise the molecules and their interactions that govern bio-adhesive systems.*
6. Line 111: 'Modern molecular biology tools' – change to 'a range of appropriate molecular biology approaches'. Some techniques are now 10-20 years plus in use.
 - 6.a. Addressed, lines 112-114 now reads: *In this study, we used a range of appropriate molecular biology approaches to investigate tidal transitory adhesion in Patella vulgata, including transcriptomics, proteomics, lectin-based assays, and in situ hybridisation.*

7. Line 115: What was 'careful' about it?
 - 7.a. We agree that 'careful' does not add value to the sentence and have revised it to be as follows (lines 115-117): *Fourteen candidate protein sequences were individually annotated with conserved protein domains, many of which are also present in published temporary adhesives from marine invertebrates.*
8. Line 165: Replace two consecutive bracketed text instances with one, using a semi-colon.
 - 8.a. Implemented.
9. Line 163-181: The circulating sea water and the ASW were of the same salinity? This isn't stated.
 - 9.a. The circulating seawater in the aquarium tank at the University of Mons was made from artificial saltwater, prepared per manufacturer's instructions (salinity reading using a refractometer was ~33‰). The same method was used to prepare syringe filtered ASW. We have revised the text to read (lines 166-168): *Limpets were allowed to settle onto thin sheets PVCA (around 200 µm thick) in an aquarium tank with circulating artificial saltwater (ASW, made per manufacturer instructions; Instant Ocean, Aquarium Systems, VA, USA) at the Biology of Marine Organisms and Biomimetics Unit, University of Mons, Belgium.*
 - 9.b. We also revised lines 179-181: *BPAM was collected directly from the pedal soles of upturned limpets by making the thin mucus film swell with a small volume of filtered ASW (Instant Ocean, per manufacturer instructions).*
10. Line 199: β -MSH = β -melanocyte-stimulating hormone in existing literature. Perhaps abbreviations, 2BME or 2 β ME are more appropriate?
 - 10.a. We thank the referee for raising this and have discarded abbreviations and used β -mercaptoethanol instead (line 204).
11. Line 318 – Authors may have over-sighted. Please add 'approximately 60%'. On measuring, it appears that pressures compared are -0.60 kPa and -0.97kPa. Not exactly -0.60 kPa and -1.00 kPa.
 - 11.a. We thank the referee for pointing this out. Lines 338-339 now reads: *This negative peak decayed slowly, taking around 7 s to reach approximately 60% of the minimum pressure.*
12. Line 319: Add perpendicularly from the surface, as written in line 310?
 - 12.a. Implemented and lines 340-342 now reads: *Normal pull off: when the limpet was allowed to settle over the sensor and then manually detached perpendicularly (arrow marks beginning of detachment), a sharp negative peak was recorded that reached -5.7 kPa, which returned to zero when the limpet detached (marked \emptyset).*
13. Line 327: I'd recommend 'protein bands' rather than 'a few proteins', cannot distinguish separate proteins from the smears, nor quantify. I see it is correctly done in line 331.
 - 13.a. Addressed. Line 357 now reads: *"protein bands larger than 250 kDa"*.
14. Line 332- Authors refer to different mucus as IPAM, BPAM, SAM. Supplementary figure 2 refers to 'old mucus, fresh mucus, footprint'. Confusing. Fix. I presume footprint is referring to IPAM?
 - 14.a. Addressed. Supplementary Figure 2 (now Supplementary Figure 3) legend and figure itself have been revised.
15. Line 347: LCA – first time this abbreviation has been stated – please write in full. Same for all lectin stains.
 - 15.a. Addressed for all lectins.
16. Lines 370-371: Did you manually identify these with knowledge and literature searches with known adhesives? Unclear how you identified. Line 374 – what 'stringent' criteria?
 - 16.a. We apologise for any confusion: the final list of 171 candidate proteins were identified based on the Methods outlined in lines 252-258 To paraphrase: peptide sequences from MS/MS were searched against all ORFs from the transcriptome using ProteinPilot. The

raw output was then filtered to remove candidates with false discovery rates above 0.01, as well as any candidates that was not present in all three individual limpet samples. The final count (minus duplicates) was 171 candidate proteins.

- 16.b. We have clarified our selection criteria (lines 406-408): *Note that, due to our selection criteria (where a candidate protein had to be present in all three limpet individuals in order to be attributed to BPAM, SAM, or IPAM), some proteins may not have been assigned to a particular type of adhesive mucus.*
17. Table 1: Don't assume readers know each lectin stain name. Full name can be provided in the legend or in the table.
- 17.a. Addressed.
18. Line 382: What database did you use for manual annotation of conserved protein domains and what search criteria was used? There are several available.
- 18.a. We used InterPro v75.0, default search parameters, as stated in Methods line 262. It is also written in the header of Table 2.
19. Line 465 – state what species lysozyme C is from. I've checked and it's *Canus lupus familiaris*. State it. Same with line 469 – *Homo sapiens*. Check throughout for others. Be consistent.
- 19.a. We are grateful that the referee has highlighted this oversight. We have added species name to the following:
- 19.a.1. Line 481-483: *P-vulgata_5 was homologous to the settlement-inducing protein complex (SIPC) from barnacles (Megabalanus coccopoma; 91% QC, 27.70% ID; BAM28692.1), both of which contain alpha-2 macroglobulin domains. This protein's homology to SIPC is discussed in the subsequent section.*
- 19.a.2. Line 498: *...lysozyme C from Canus lupus familiaris (88% QC, 38.03% ID; NP_001300804.1).*
20. Line 625 –The following statement is quite bold– “*P. vulgata_14* is the first example of an annotated protein from marine bio-adhesive with multiple domains relevant for antibacterial activity”. Several papers on echinoderms, anemones and others highlight proteins potentially associated with antimicrobial activity / immunity / defence. Perhaps change the wording as wet-lab verification work needs completed to assign a function to the protein. It well could have a different role?
- 20.a. We agree with the reviewer that additional experiments are required to verify our claims; hence, we have removed the sentence and revised the text as follows (lines 667-674): *The abundant target recognition and regulation-related domains suggest that P-vulgata_14 acts as an antibacterial agent within the pedal mucus. Interestingly, since this protein was found only in the BPAM samples, possible roles of P-vulgata_14 are to: (1) prevent microbial degradation of the secreted mucus, which has to remain functional over prolonged periods of time (e.g., during high tide, when the limpet typically stops foraging and remains stationary within the safety of its home scar), and (2) to minimise risk of infection. However, many aspects of this protein need to be further investigated to verify its purported function, such as its target specificity, stability, and how it interacts with the gel network.*
21. Line 685-686: No tyrosinase like orthologs within the transcriptome? If not, you could add this to this section.
- 21.a. We found several instances of predicted tyrosinase-like orthologs from BLAST search of the pedal sole transcriptome. Examples include tyr-3 ortholog from *Aplysia californica* and *Mizuhopecten yessoensis*. None were detected from the 171 MS/MS matched protein candidates.
22. Supplementary Table 2 – there is a bracket in the legend, remove.
- 22.a. Addressed.
23. Supplementary table 3: What does NA mean? – I presume these are the ones that failed in ISH?

- 23.a. We agree that this was unclear and have added the following clarification beneath Supplementary Table 3: **: Not applicable (NA), where a manuscript ID was not generated for a given Trinity-assigned protein ID as it was not included in downstream manual annotation and ISH was unsuccessful.*
24. Supplementary figure 3: legend and alignments on one page please.
- 24.a. Addressed.
25. Supplementary: “DNA sequences of the fourteen annotated protein” – please change to “Encoding cDNA” as RNA-seq captures RNA which is then reverse transcribed to cDNA for sequencing, not gDNA which also includes introns – I assume a simple over-assumption by authors. Line 762 as well.
- 25.a. We thank the reviewer for the correction; which we have now implemented.
26. Additional data: Please publish the assembled transcriptome with raw reads to NCBI. It will be a good resource for the bio-adhesive community.
- 26.a. We have uploaded the raw reads and assembled transcriptome to NCBI BioProject PRJNA613775. Our submission is currently being processed and will be made available upon publication.

I look forward to receiving the resubmitted version.

Referee 2

We would like to thank Referee 2 providing insightful feedback and helpful suggestions. Please see our response to individual points below.

Minor comments:

1. There are large differences in the abundance of the different transcripts based on the FPKM data. This seems very important and should be discussed. The relative abundance gives some insight into the role of the proteins. I would expect something that is extremely abundant to be more likely to be a major structural component of the secretion, whereas something that is much less abundant may serve a more specialized role, or a catalytic role.
 - 1.a. We thank the reviewer for this suggestion. Although we agree that transcript expression levels may offer additional insight into the potential role of the proteins, we would like to refrain from discussing this at length in our manuscript for the following reasons. First, since our transcriptome captures a single time-point of expression (at the moment of sample preparation) from a single individual, we do not believe the FPKM numbers accurately represent the complex temporal dynamics of the different mucus types secreted by the limpet. Second, it is difficult to draw conclusions about protein abundance based purely on FPKM values (Liu Y, Beyer A, Aebersold R. *Cell*. 2016;165:535–50). We have included FPKM in Table 2 to give the readers an idea of mRNA expression levels in case that information is useful for their particular query. We hope that future studies will leverage our findings to select target proteins for quantification to understand when, where (using ISH), and how much of the adhesive proteins are produced. An example technique would be exponentially modified protein abundance index (emPAI), which has been used previously to calculate the relative abundance of sea star foot protein-1 (SFP-1) in *Asterias rubens* (Hennebert *et al.* *Proc Natl Acad Sci USA*. 2014;111:6317–22). We have added a few sentences to the Discussion that mentions these points in the context of *P-vulgata_3* (lines 724-729): *However, more work is needed to understand the role of P-vulgata_3, including its relative abundance within the different types of adhesive mucus. Although its transcript expression level was low (Table 2), it is difficult to draw conclusions about protein abundance based solely on the transcript expression levels, especially when the FPKM is derived from a single individual transcriptome. Follow-up studies using*

techniques like exponentially modified protein abundance index (emPAI) can provide quantitative information on the abundance of P-vulgata_3 and other adhesive proteins.

2. Related to the previous point, what were the criteria for selecting the fourteen sequences for more detailed analysis? Some seem to be very low abundance, based on the FPKM data, and some lack signal peptides suggesting that they aren't secreted. Without knowing the selection criteria, it is hard to be confident that all the main proteins in the glue are represented in the fourteen selected. It does seem that the authors succeeded in capturing the main proteins, since some of the most abundant transcripts from among the fourteen do have sizes that might line up with the proteins seen in SDS-PAGE. Nevertheless, more information on the selection criteria would help.
 - 2.a. We acknowledge that more information on the ISH selection criteria is necessary, and have revised the main text to read (line 270-277): *From the combined list of candidate proteins, a subset was selected for further analysis using in situ hybridisation (ISH) based on the following criteria: first, we limited our selection to proteins that were ranked highly by ProteinPilot to ensure we were targeting proteins that were present in adhesive mucus. Second, we included candidates with conserved protein domains that were commonly associated with marine bio-adhesives (e.g., vWFD, EGF, lectins). Finally, we sought to sample proteins across the different types of mucus with the goal of identifying candidates associated with specific types of mucus (IPAM, BPAM, and SAM).*
 - 2.b. Please note that we did not consider FPKM during the selection of ISH candidates since this only provides a snapshot of expression activity at the moment of sample preparation. Moreover, relying on expression levels alone could have caused us to miss proteins like *P-vulgata_3*, which had a low FPKM but was well-represented in the proteome (based on ProteinPilot ranking) and resulted in a successful ISH.
3. How was the pressure sensor calibrated? The authors should provide evidence that the measurements are accurate.
 - 3.a. We thank the referee for highlighting this, and we have added details on the calibration in the Supplementary Materials (pg 9) and have referred to it in the main text (line 141-142).
4. Line 54 should be qualified. Instead of "the mechanisms responsible for the limpets' strong attachment remain unresolved", it should read "the mechanisms responsible for patellid limpets' strong attachment remain unresolved". As the authors make clear in their review of the literature, the mechanism for lottiid limpets seems relatively well-resolved.
 - 4.a. We thank the reviewer for this suggestion and have revised accordingly (line 54-55): *Despite over a century of research, the mechanisms responsible for the patellid limpets' strong attachment remain unresolved [4].*
5. In Fig 3c, it would be helpful to know when the force was applied, and when the limpet detached. Did the sudden drop in pressure correspond with a sudden increase in detaching force, or was the force applied for some time and the pressure only dropped at the moment of failure? The latter would be further evidence of gluing. It might be worth noting that a glued animal might produce the sudden pressure drop after adhesion fails and the foot begins to deform and pull away. In Smith's paper on lottiid limpets, it was noted that this effect can create a transient pressure drop of about 8 kPa. This is comparable to what the authors see in Fig 3c.
 - 5.a. We thank the reviewer for this suggestion and have modified Figure 3c to show when the force was applied and when the limpet was detached.
 - 5.b. We also provide more detail regarding Figure 3c in the Supplementary Materials (pg 10) and refer to it in the main text (line 334).
 - 5.c. Note: the negative peak was misreported as -5.3 kPa and has been rectified to be -5.7 kPa in the revised manuscript.
6. In Supplementary Figure 2, the authors refer to footprint mucus and adhesive mucus. They should use their terminology of IPAM, BPAM and SAM throughout the paper.
 - 6.a. Addressed.

7. On line 260, the authors state that samples were fixed in paraformaldehyde. Were they then embedded in paraffin?
- 7.a. We thank the reviewer for highlighting this omission and have revised the text as follows (line 282-284): *Limpet tissues were fixed in 4% paraformaldehyde (PFA) in PBS, embedded in paraffin wax, then sectioned into 14 μm sections using a Microm HM 340 E microtome.*
- 7.b. As Referee 1 also mentioned, the initial ISH protocol provided in the Methods was too simplistic and insufficient to ensure reproducibility. Thus, we have included the full ISH protocol in Supplementary Materials (pg 8-9).

Referee 3

We thank the referee for the positive comments.